# Praja2 controls P-body assembly and translation in glioblastoma by non-proteolytic ubiquitylation of DDX6

Emanuela Senatore[1], Rosario Avolio[1], Laura Rinaldi[1], Francesco Chiuso[1], Maria A Oliva [ID][2], Chiara D'Ambrosio [ID][3], Antonio Giuseppe Bianco[1], Emiliano Dalla[4], Stefano Maria Pagnotta [ID][5], Raffaella Flammia [ID][5], Concetta Ambrosino[5], Domenico Memoli [ID][6], Gabriele Turacchio[7], Sonia Ines Mimoune[8], Yves Toiron[8], Stephane Audebert [ID][8], Luc Camoin [ID][8], Luca Lignitto [ID][8], Andrea Scaloni [ID][3], Antonietta Arcella[2] & Antonio Feliciello [ID][1✉]

## Abstract

**Glioblastoma multiforme (GBM) is the most lethal form of malignant brain tumor in adults. Dysregulation of protein synthesis contributes to cancer cell plasticity, driving GBM cell heterogeneity, metastatic behavior, and drug resistance. Understanding the complex network and signaling pathways governing protein translation, is therefore an important goal for GBM treatment. Here we identify a novel signaling network centered on the E3 ubiquitin ligase praja2 that controls protein translation in GBM. Praja2 forms a multimeric complex with the RNA helicase DDX6, which inhibits translation of target RNAs within processing bodies (P-bodies). Stimulation of cAMP signaling through activation of G-protein-coupled receptors induces P-body assembly through praja2-mediated non-proteolytic polyubiquitylation of DDX6. Genetic inactivation of praja2 reshapes DDX6/mRNA complexes and translating polysomes and promotes cellular senescence and GBM growth arrest. Expression of an ubiquitylation-defective DDX6 mutant suppresses the assembly of P-bodies and sustains GBM growth. Taken together, our findings identify a cAMP-driven network that controls translation in P-bodies and GBM growth.**

**Keywords** Praja2; cAMP; PKA; P-body; Glioblastoma
**Subject Categories** Cancer; Post-translational Modifications & Proteolysis; RNA Biology

## Introduction

Glioblastoma (GBM) is the most lethal form of malignant brain tumor in the adult population representing about 12–15% of all intracranial neoplasms and 60–75% of glial tumors, with an incidence of 3–4 cases every 100,000 individuals and 5-years survival rate of less than 5% (Schaff and Mellinghoff, 2023). The standard therapy for patients with newly diagnosed GBM includes tumor resection surgeries, followed by radiotherapy and chemotherapy (Stupp et al, 2005). A detailed molecular map of GBM lesions integrated with histologic features emerged as an important predictive value for the prognosis and the limited therapeutic options for surviving patients. Specifically, mutations of isocitrate dehydrogenase 1 gene (IDH1) and co-deletion of chromosomes 1p-19q are determining factors to delineate different histomolecular GBM subtypes. The concomitant presence of IDH1 mutation and 1p-19q co-deletion are classified as oligodendroglial variants, whereas neoplasms carrying IDH1 gene mutation in the absence of 1p-19q co-deletion are assigned to astrocytic forms, as well as tumor lesions with no evidence of IDH1 mutations (Prager et al, 2020; Van Den Bent et al, 2017; Caccese et al, 2020; Trautwein et al, 2022). GBM is known to relapse, making treatment difficult (Lo Greco et al, 2022). Thus, this tumor urgently demands novel therapeutic strategies aimed to ameliorate the quality of life and survival rate of patients.

An important aspect concerning the malignant behavior of GBM is the peculiar versatility to rapidly adapt to changes in metabolic needs, hypoxia, hyperosmolarity and nutrients availability. The high proliferative rate often linked to dysregulation of protein translation as consequence of protein misfolding, unbalanced protein synthesis and sorting within secretory pathways, contributes to generate a hostile microenvironment that would otherwise affect growth and dissemination (De Silva et al, 2023; Lee et al, 2021). The dynamic and rapid control of protein translation,

[1]Department of Molecular Medicine and Medical Biotechnology, University Federico II, Naples, Italy. [2]I.R.C.C.S Neuromed, Pozzilli, Isernia, Italy. [3]Proteomics, Metabolomics and Mass Spectrometry Laboratory, ISPAAM, National Research Council, Portici, Naples, Italy. [4]Department of Medicine, University of Udine, Udine, Italy. [5]Department of Science and Technology, University of Sannio, Benevento, Italy. [6]Department of Medicine, Surgery and Dentistry SMS, University of Salerno, Salerno, Italy. [7]Institute of Biochemistry and Cell Biology, National Research Council, Naples, Italy. [8]Cancer Research Center of Marseille (CRCM), CNRS, Aix Marseille University, INSERM, Institut Paoli-Calmettes, Marseille, France. ✉E-mail: feliciel@unina.it

thus, represents a fundamental mechanism that cancer cells adopt to temporarily exit cell cycle, surviving in presence of restrictive growth conditions (Montiel-Dávalos et al, 2023). Maturation and nuclear export of mRNAs are linked to protein translation. However, before their transport to other cellular compartments or in the presence of stress conditions, serum deprivation or changes in metabolic needs, mRNAs can be temporarily stored within non-membranous ribonucleoprotein cytoplasmic condensates, namely processing bodies (P-bodies), that form through a dynamic process of liquid-liquid phase separation (LLPS) (Luo et al, 2018). Stored mRNAs may undergo translational repression, RNA-mediated gene silencing or decay, but can also resume translation (Parker and Sheth, 2007). A vast collection of proteins has been identified as components of P-bodies, including CCR4-Not deadenylating complex, mRNA Lsm1-7 decay complex, decapping enzymes and activators (DCP1/DCP2, EDC3/EDC4 and PAT1), the RNA helicase DDX6 and the ribonuclease XRN1 (Parker and Sheth, 2007; Hubstenberger et al, 2017; Youn et al, 2018). Moreover, other proteins responsible for translational repression, such as 4E-T and CPEB3, have been identified as components of P-bodies (Kamenska et al, 2014; Ford et al, 2019). Despite the large number of proteins found within P-bodies, only DDX6, EDC3 and Lsm4 are essential for the formation of these membrane-less organelles (Ayache et al, 2015; Kamenska et al, 2016; Decker et al, 2007).

Signaling pathways play a major role in P-body dynamics and different protein kinases have been functionally associated with the regulation of P-body assembly/disassembly (Rzeczkowski et al, 2011; Bearss et al, 2021). In particular, c-Jun-N-terminal kinase (JNK) phosphorylates DCP1A at serine315 determining its delocalization from P-bodies and affecting P-body number (Rzeczkowski et al, 2011). The kinase Pim1 and AKT phosphorylate EDC3 and reduce its localization at P-bodies, affecting membrane-less organelles assembly (Bearss et al, 2021). In yeast, carbon source deprivation induces RNPs aggregates within P-bodies that repress protein translation and support long-term survival of cells at the stationary phase. Glucose supplementation dissolves P-bodies and restore protein translation and cell growth. This mechanism depends on PKA-mediated phosphorylation of Pat1, a conserved component and assembly factor of P-bodies. Pat1 phosphorylation prevents the recruitment of Dhh1 (human protein ortholog of DDX6) at nucleating P-bodies. The inactivation of PKA, but not of other nutrients sensing kinases, resumes P-bodies and represses translation (Ramachandran et al, 2011).

Emerging evidence indicates that also the ubiquitin system may impact assembly of P-bodies and mRNA translation (Tenekeci et al, 2016; Wei et al, 2022; Kedia et al, 2022). Thus, TRAF6 mediated ubiquitylation of DCP1A promotes the formation of the decapping complex with mRNA decay factors DCP2, EDC4 and Xrn1, and P-body remodeling (Tenekeci et al, 2016). The binding between DCP1A and EDC4 is also modulated by the Trim24-mediated ubiquitylation of EDC4, that inhibits the decapping complex formation, thus regulating the stability of specific mRNAs via P-bodies (Wei et al, 2022). Moreover, the ubiquitylation state of 4E-T, controlled by the E3 ligase Trim56 and the deubiquitinase Otud4, regulates P-body dynamics (Kedia et al, 2022). Praja2 is a mammalian E3 ubiquitin ligase that efficiently couples phosphorylation to ubiquitylation of protein kinases, scaffolds and downstream effectors, playing a major role in cell signaling, metabolic

pathways and development (Lignitto et al, 2011, 2013; Sepe et al, 2014; Rinaldi et al, 2016; Gong et al, 2020; Zhao et al, 2021; Zhong et al, 2017; Senatore et al, 2021; Chiuso et al, 2023; Rinaldi et al, 2024; Schiefer and Hale, 2024). Deregulation of praja2-controlled signaling pathways are mechanistically linked to the development and progression of human glioblastoma (Lignitto et al, 2013; Delle Donne et al, 2022). Recently, a critical role of P-bodies in tumorigenesis and cancer dissemination emerged. In breast cancer cells, P-bodies control key steps of epithelial–mesenchymal transition (EMT) and distal metastasis (Hardy et al, 2017). In colon cancer cells, oncogenes YAP/TAZ promote P-bodies assembly through a transcriptional program that increases the expression of core components of the organelles. This mechanism underlies cancer cell proliferation, migration and colon cancer growth (Shen et al, 2024). However, in most human cancers, including GBM, the intricate link between PKA, protein ubiquitylation and P-body regulation and the impact of dynamics of P-bodies in tumorigenesis and metastasis are largely unknown.

Here, we report the existence of a cAMP-regulated ribonucleoprotein network controlled by the RING ligase praja2 that finely regulates P-body assembly and mRNA translation in GBM. In response to cAMP stimulation, praja2 ubiquitylates DDX6 and promotes the assembly of P-bodies, significantly impacting on DDX6/mRNA complex formation and protein translation. Interfering with this regulatory circuitry promotes cellular senescence and GBM growth arrest.

## Results

### Role of praja2 in the transcriptional landscape of GBM

Given the role of praja2 in the mitogenic signaling and metabolic pathways, we evaluated the impact of praja2 downregulation on the transcriptional reprogramming of GBM cells. To this end, we performed RNA sequencing analysis in GBM cells (U87MG) transiently transfected with siRNAs targeting endogenous praja2 (sipraja2) or with control siRNAs (siCNT). We identified 668 differentially expressed (DE) transcripts in praja2-silenced cells, compared to controls. In particular, we found that 340 mRNAs were upregulated in the praja2-silenced group, while 328 transcripts, including praja2 mRNA, were downregulated. The False Discovery Rate (FDR) has been controlled with the Benjamini and Hochberg correction of $p$ values, resulting in the rule FDR < 0.01 (Fig. 1A). We also generated a heatmap showing the most differentially expressed genes between both conditions (Fig. 1B). To understand the molecular significance of the transcriptional changes induced by praja2, we performed an enrichment analysis by querying differentially expressed transcripts with gene sets from the collections Hallmark, Biological Processes, and Molecular Functions of MSigDB (Liberzon et al, 2011). We obtained a network of significant gene sets with FDR < 0.01, in which the size of circles was associated with the size of the corresponding gene set and the intensity of the color correlated with the magnitude of the enrichment score (Fig. 1C). The results show that transcripts enriched in sipraja2 cells (red circles) were related to cell adhesion and migration, neurogenesis, cellular response to bone morphogenetic pathway, regulation of ion transport, synaptic transmission, and oxidative phosphorylation. In contrast, transcripts involved in

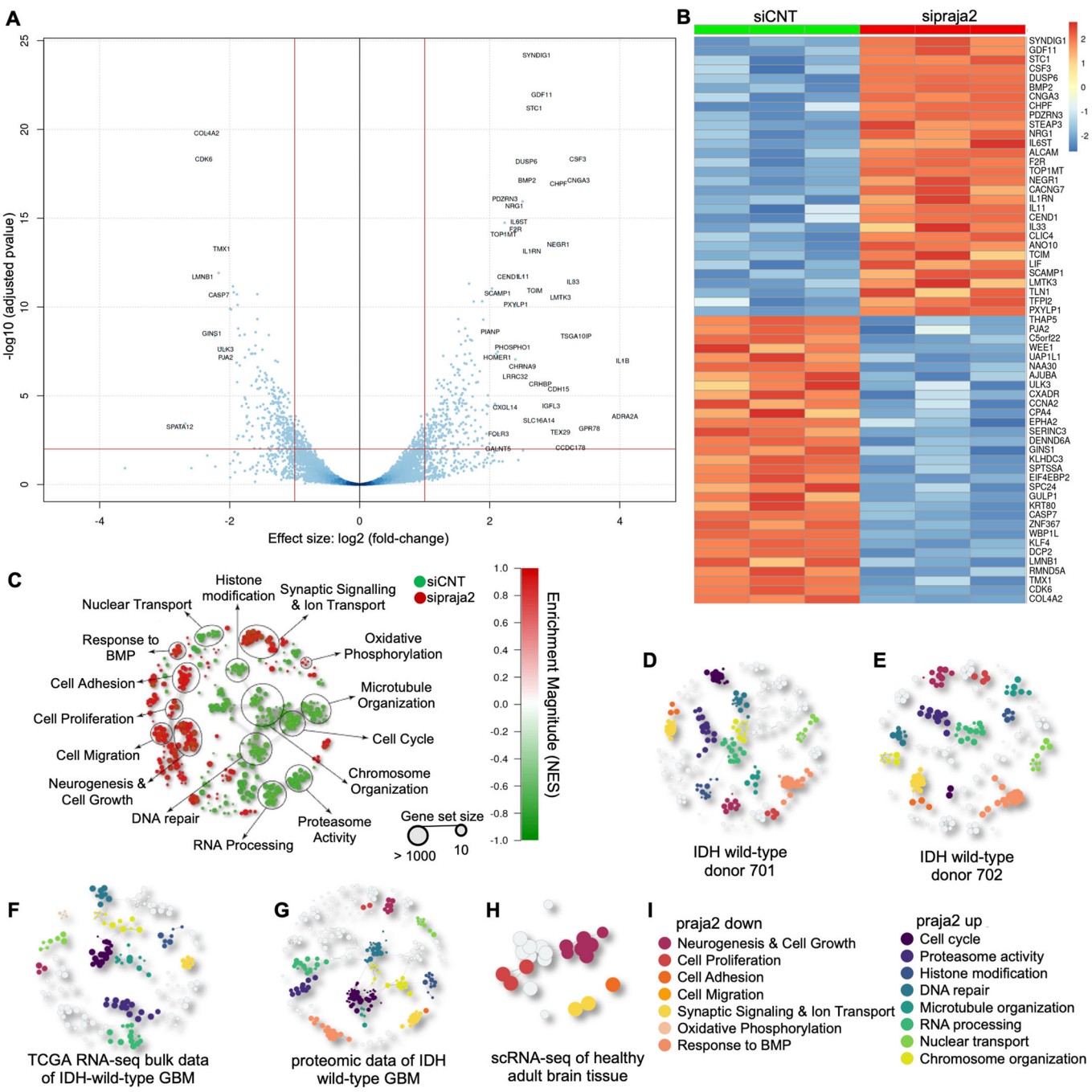

**Figure 1. The role of praja2 in the transcriptional landscape of GBM.**

(A) Volcano plot showing differentially expressed transcripts comparing RNAseq data from U87MG cells transfected with siRNA targeting endogenous praja2 (sipraja2) to negative controls (siCNT) (n = 3 biological independent replicates). FDR < 0.01 has been considered. The statistical test used for the analysis of RNAseq data was Wald test as implemented in the R package DeSeq2. (B) Heatmap of the most differentially expressed genes between sipraja2 and siCNT U87MG cells. Shown are 30 transcripts up- or downregulated. (C) Network representation of statistically significant gene sets (FDR < 0.01) enriched in siCNT (green circles) or sipraja2 (red circles) U87MG cells. Each dot displays a gene set from the Hallmark collection or GO terms from the Biological Process or Molecular Functions category. The circle size is proportional to the number of items in the gene set, and the intensity of colors is proportional to the magnitude of the enrichment score. (D–H) Network representations of in silico enrichment analysis showing the correlation between praja2 expression levels and gene transcription profile of different GBM tissues dataset. Colored circles represent statistically significant gene sets (FDR < 0.01) with a significant intersection with those in (C). (D, E) are scRNA-seq datasets of two donors (701 and 702 in GSE129671 (Data ref: Ding et al, 2019)) of IDH wild-type tissue. (F) is the bulk RNAseq study of IDH wild-type GBM in The Cancer Genome Atlas repository (Data ref: TCGA-GBM; phs000178). (G) is a proteomic array of IDH wild-type GBM (PDC000204) (Data ref: Wang et al, 2021); (H) is a scRNA-seq dataset of healthy adult brain tissue (GSE67835) (Data ref: Darmanis et al, 2015). (I) Legends of macrocategories showed in (D–H).

cell cycle, proteasome activity, RNA processing, DNA repair, and histone modification were enriched in siCNT cells, compared to sipraja2 cells (green circles) (Fig. 1C). To further prove that praja2 had an impact on the transcriptome of GBM cells, we performed a correlation analysis between expression levels of praja2 and the enrichment of specific gene sets in human GBM tissues. To this end, we performed an in silico enrichment analysis on data from different datasets of GBM samples in which we evaluated the Pearson linear correlation index between praja2 expression levels and the gene transcription profile (Fig. 1D–I). In particular, we considered datasets from two IDH wild-type donors of scRNA-Seq (GSE129671; (Ding et al, 2019; Data ref: Ding et al, 2019) (Fig. 1D,E), bulk RNAseq data of IDH wild-type samples from The Cancer Genome Atlas study of GBM (Data ref: TCGA-GBM; phs000178) (Fig. 1F), proteomic data of IDH wild-type GBM (PDC000204; (Wang et al, 2021; Data ref: Wang et al, 2021) (Fig. 1G) and scRNA-seq data of healthy adult brain tissue (GSE67835; (Darmanis et al, 2015; Data ref: Darmanis et al, 2015) as control (Fig. 1H). We selected significant gene sets with FDR < 0.01 and the results demonstrated a significant correlation between the levels of praja2 transcript and the enrichment of different pathways (Fig. 1D–I; Dataset EV1). Next, we compared each in silico enrichment analysis of different datasets with the enrichment analysis performed in GBM cells (shown in Fig. 1C), and we found that the same pathways were differentially enriched (indicated as colored circles) (Fig. 1D–I). The intersection was statistically significant (for more detailed information, see "Methods"). In particular, pathways that in tissues gene sets correlated with low levels of praja2 (praja2 down) were also found enriched in praja2-silenced cells (Fig. 1C, red circles). Similarly, pathways that correlated with high levels of praja2 in tissues gene sets (praja2 up) were also found enriched in control U87MG cells (Fig. 1C, green circles). Conversely, as expected, we found a poor intersection between U87MG cells and datasets from normal brain tissue (Fig. 1H). Altogether, the results support a role of praja2 in the regulation of the transcriptional landscape in GBM.

## Proteomic analysis identified the RNA helicase DDX6 as a novel target of praja2

To identify praja2 interactors that could explain its involvement in the regulation of the transcriptional reprogramming, we evaluated the proteomic analysis of affinity-purified praja2 complexes from total cell lysates (Delle Donne et al, 2022; Rinaldi et al, 2024) (Dataset EV2). We retrieved direct protein–protein interactions (PPIs) using the inBio Discover and STRING webtools, based on highly trusted interactions derived from experimental evidence. The undirected PPI network, representing the interactome of praja2, was constituted of 506 interconnected and 215 non-connected proteins, with three interactors not recognized by the tool (Fig. EV1). To characterize the main biological functions associated with praja2 interactors, we performed a functional enrichment analysis on the generated PPI network. First, we examined the top 5 most enriched REACTOME terms. The subnetwork composed of 237 interactors was strikingly associated with RNA and protein metabolism ($n = 95$ and $n = 132$ interactors, respectively) and to a lesser extent with cell cycle and stress response (Fig. EV2). We specifically focused on the subnetwork involved in RNA metabolism and processing. The STRING analysis identified a significant number of proteins ($n = 107$) involved in functions related to mRNA, rRNA,

ncRNAs and P-bodies (Fig. 2A,B; Dataset EV3). This analysis was supported by the identification through yeast two-hybrid screening of RNA helicase DDX6 as a putative praja2-binding partner (Lignitto et al, 2011). DDX6 is a family member of DEAD box genes localized at P-bodies, where it plays a major role in translational repression and mRNA turnover (Luo et al, 2018; Decker and Parker, 2012). Moreover, we performed an unsupervised clustering that aggregated cells from the IDH wild-type GBM patient donor 701 (GSE129671, (preprint: Ding et al, 2019; Data ref: Ding et al, 2019)) (Fig. EV3A). The hierarchical method, paired with Ward's linkage applied to a Euclidean matrix, allowed the aggregation of more than 20.000 cells (AWST protocol (Risso and Pagnotta, 2021), as shown in (Pagnotta, 2024). The analysis identified four clusters associated with different expression levels of praja2. In particular, we found that the transcripts levels of different P-bodies components (DDX6, DCP1A, DCP1B, and EDC3) followed a pattern similar to praja2 mRNA, as shown by bar plots indicating averaged expression levels of each transcript in the identified clusters (Fig. EV3B). These data supported the concept that praja2 is a component of protein complexes assembled at P-bodies and a potential regulator of mRNAs turnover/translation.

## Molecular characterization of praja2-DDX6 complex

By immunogold electron microscopy (EM) analysis, we found that praja2 partly localized in occasional electron-dense membrane-less structures occurring in the cytoplasm and having a diameter of about 100 nm. The size, the number and intracellular distribution of immunogold structures suggested that praja2 might be a component of P-bodies (Fig. 3A). As predicted, in situ immunostaining analysis, both in HEK293 and U87MG cell lines, confirmed that a portion of praja2 signal partially colocalizes with DDX6 at P-bodies (Fig. 3B,C). Co-immunoprecipitation experiments confirmed the existence of an endogenous praja2/DDX6 protein complex in cell lysates (Fig. 3D). Similarly, overexpressed epitope-tagged DDX6 and praja2 form a stable complex in lysates (Fig. 3E). Furthermore, we found that praja2 co-purified with other components of P-bodies involved in translational repression, mRNA decapping, and RNA degradation, specifically, decapping mRNA 1A (DCP1A) and enhancer of mRNA decapping 3 (EDC3) (Seto et al, 2015; Tritschler et al, 2007) (Fig. 3E,F). To assess the molecular determinants mediating the interaction between praja2 and DDX6, we mapped the praja2-binding site for DDX6 using praja2 deletion mutants (Fig. 3G). The data reported in Fig. 3H indicated that the praja2 domain spanning residues 530–630 is required for binding to DDX6. Moreover, GST-pull-down experiments demonstrated that a purified GST-praja2 polypeptide (residues 530–630) is sufficient to directly bind DDX6 (Fig. 3I). This is a conserved domain of praja2 that forms a long α-helix structure that mediates the interaction also with other praja2 targets (Lignitto et al, 2011, 2013; Senatore et al, 2021; Chiuso et al, 2023).

## DDX6 is a direct target of the ubiquitin pathway

Next, we evaluated if praja2 ubiquitylates DDX6. As shown in Fig. 4A, wild-type praja2 overexpression, but not its inactive RING mutant (praja2rm) (Senatore et al, 2021), promoted a marked polyubiquitylation of the co-expressed DDX6. Praja2 activity is induced by PKA-mediated phosphorylation at serine 342 and threonine 389. To test if PKA phosphorylation of praja2 was required to induce DDX6 ubiquitylation, we analyzed DDX6

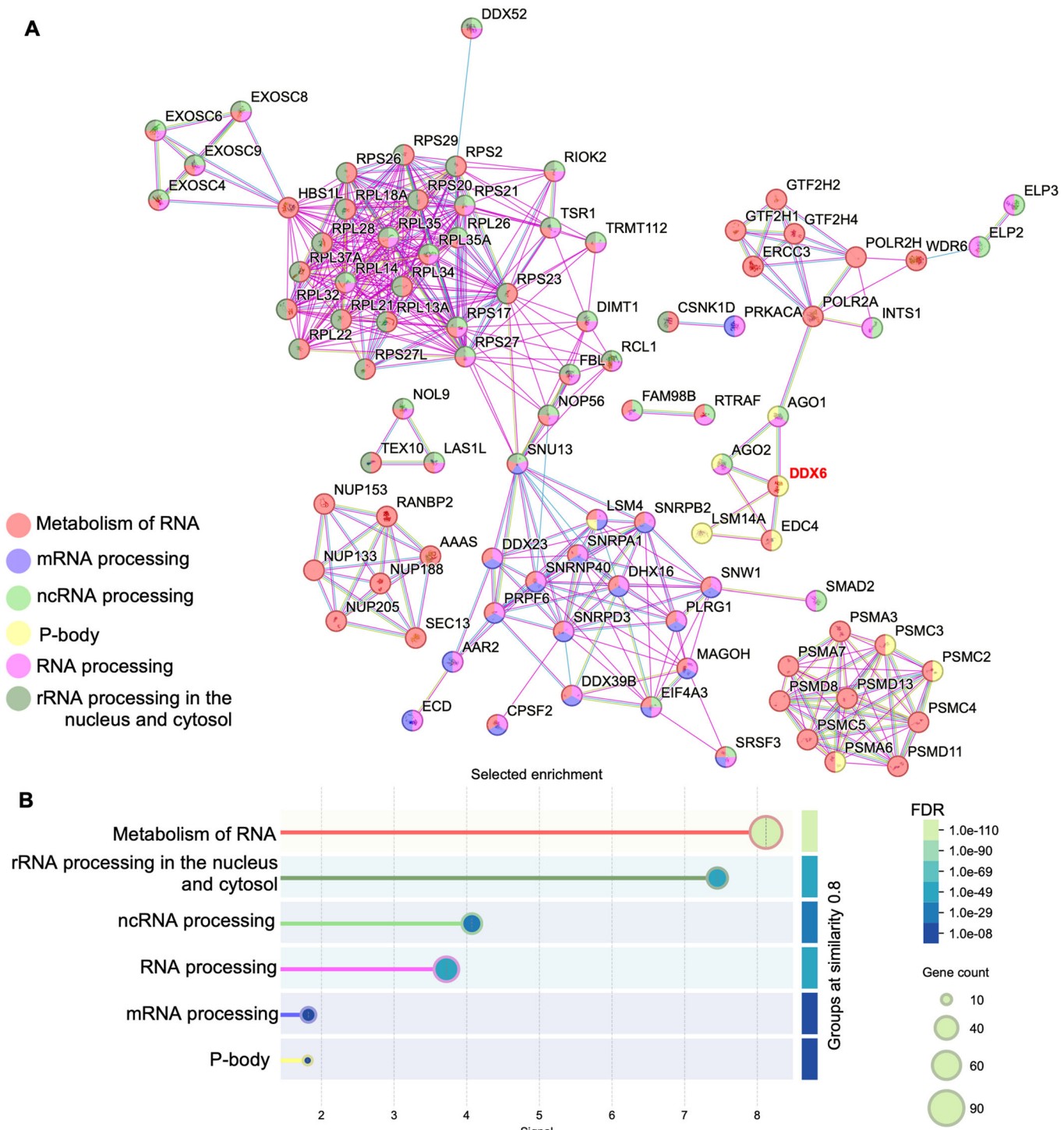

**Figure 2. Identification of DDX6 as new interactor of praja2.**

(A) Protein–protein interaction network of components involved in RNA metabolism and processing. The STRING web tool was used to query the Gene Ontology and Reactome databases for functional terms enrichment. The interactions existing between proteins involved in metabolism of RNA R-HAS-8953854 (red nodes), mRNA processing GO:0006397 (blue nodes), ncRNA processing GO:0034470 (green nodes), RNA processing GO:0006396 (pink nodes), rRNA processing in the nucleus and cytosol R-HAS-8868773 (olive green nodes) and P-bodies GO:000932 (yellow nodes) are shown (n = 107 proteins, n = 454 interactions). (B) Detailed view of the six selected enriched functional terms described in (A). The STRING web tool was used to generate a plot of enriched functional terms (Group terms by similarity >=0.8) from the Gene Ontology and Reactome databases. The signal (x axis) represents the weighted harmonic mean between the observed/expected ratio and −log(FDR) of each functional term; node size is proportional to the number of genes associated with each term while node filling colors indicate the statistical significance (FDR) of enriched terms.

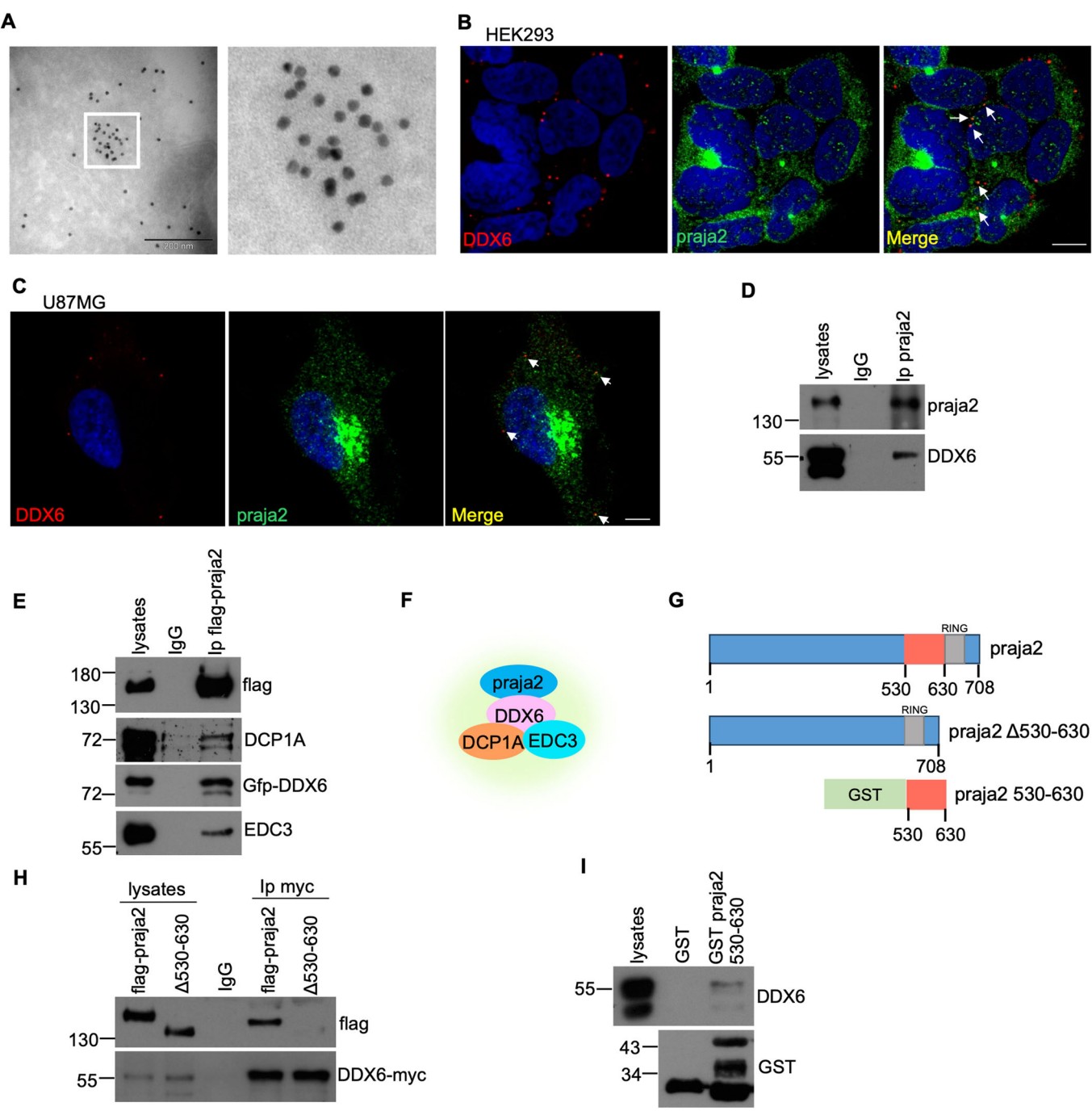

**Figure 3. Praja2 binds to and colocalizes with DDX6 at P-bodies.**

(A) Electron microscopy of HeLa cells incubated with praja2 antibody. Scale bar: 200 nm. (B) Staining of HEK293 cells with anti-praja2, anti-DDX6 and DAPI. Scale bar: 5 µm. (C) Staining of U87MG cells with anti-praja2 and anti-DDX6 antibodies, and DAPI. Scale bar: 5 µm. (D) U87MG lysates were immunoprecipitated with anti-praja2. Lysates and immunoprecipitates were immunoblotted with anti-praja2 and anti-DDX6. (E) HEK293 cells were co-transfected with flag-praja2 and Gfp-DDX6. Cell lysates were immunoprecipitated with anti-flag. Lysates and immunoprecipitates were immunoblotted with anti-flag, anti-Gfp, anti-DCP1A and anti-EDC3. (F) Schematic picture of the praja2-DDX6-DCP1A-EDC3 complex. (G) Schematic model representing the deletion mutant of praja2 used. (H) HEK293 cells were co-transfected with flag-praja2 (wild-type and Δ530–630) and DDX6-myc. Cell lysates were immunoprecipitated with anti-myc. Lysates and precipitates were immunoblotted with anti-myc and anti-flag. (I) HEK293 cells were subjected to pull-down with GST and GST fused to amino acid 530–630 of praja2. Precipitates and lysates were immunoblotted with anti-GST and anti-DDX6. Source data are available online for this figure.

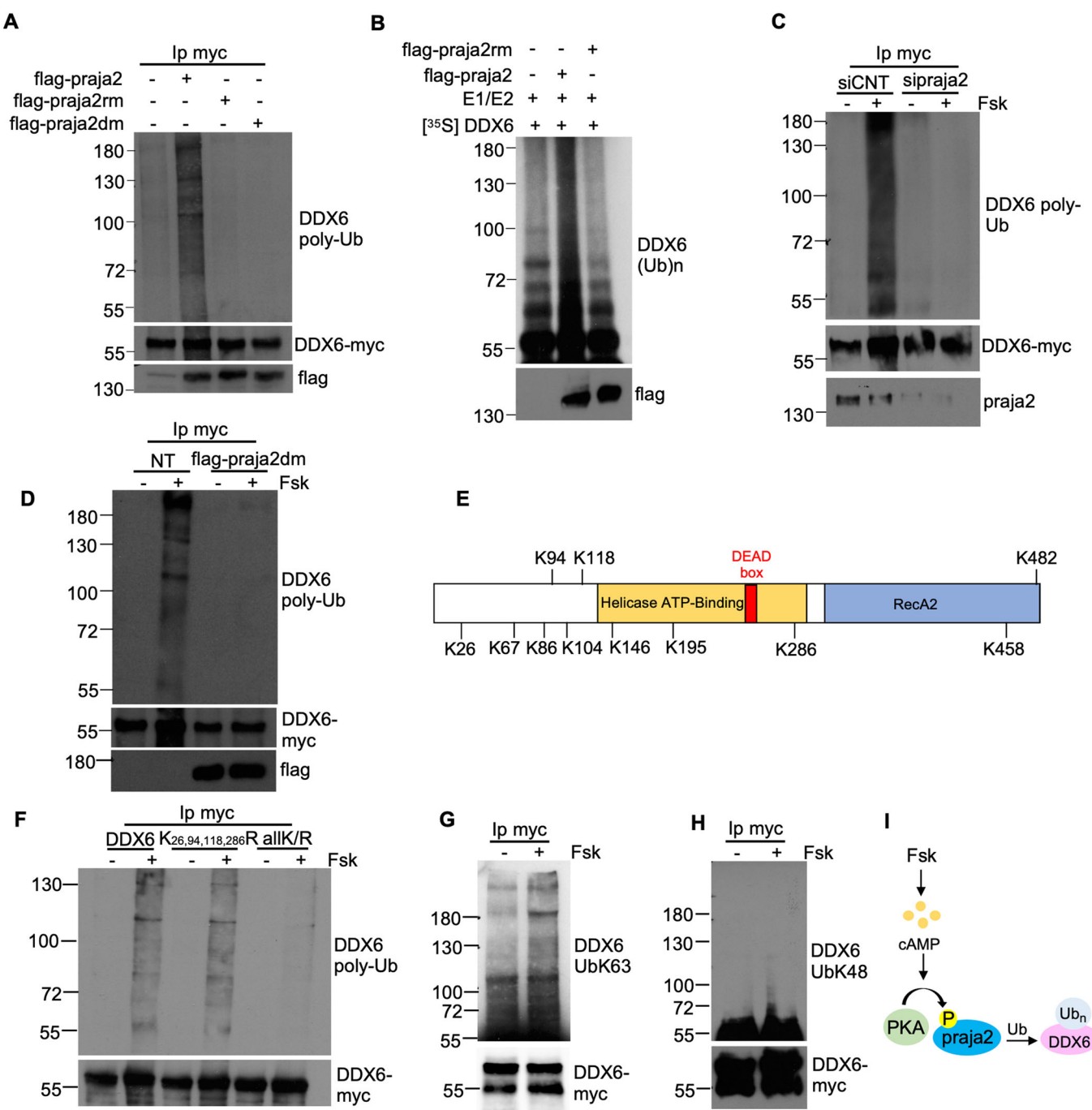

ubiquitylation in cells expressing a phosphorylation-deficient praja2 mutant carrying substitution of S342/T389 with alanine (praja2dm) (Chiuso et al, 2023). Figure 4A shows that over-expression of praja2dm, similarly to praja2rm, was not able to induce ubiquitylation of DDX6, compared to wild-type protein. Ubiquitylation experiments using in vitro translated, $^{35}$S-labeled recombinant DDX6 demonstrated that immunopurified wild-type praja2, but not praja2rm, directly ubiquitylates DDX6 (Fig. 4B). Since praja2 activity is induced by protein kinase A (PKA), we tested if DDX6 ubiquitylation was regulated by cAMP. To this aim,

we evaluated the ubiquitylation pattern of DDX6 in cells treated with Forskolin (Fsk), a natural diterpene that activates adenylyl cyclase and increases cAMP levels. The results shown in Fig. 4C demonstrated that Fsk treatment markedly increased the poly-ubiquitylation of DDX6, which in turn was mostly abrogated by genetic silencing of praja2. As expected, overexpression of praja2dm completely abrogated DDX6 ubiquitylation induced by Fsk treatment, confirming that cAMP-PKA-praja2 axis regulates DDX6 polyubiquitylation (Fig. 4D). Next, we attempted to identify the lysine residue(s) of DDX6 that accept ubiquitin molecules

**Figure 4. cAMP induces the praja2-mediated non-proteolytic ubiquitylation of DDX6.**

(A) HEK293 cells were co-transfected with DDX6-myc, Ubiquitin-HA, and flag-praja2 wild-type, ring mutant (praja2rm) or phosphorylation mutant (praja2dm). Lysates were immunoprecipitated with anti-myc. Precipitates were immunoblotted with anti-HA (DDX6 poly-Ub) and anti-myc, while lysates were immunoblotted with anti-flag. (B) In vitro ubiquitylation assay of in vitro translated $^{35}$S-labeled DDX6 in the presence of flag-praja2 or flag-praja2rm precipitates, his$_6$-tagged ubiquitin, E1, and Ubch5b enzymes. The reaction mixture was analyzed by autoradiography. An aliquot of the precipitates was immunoblotted with anti-flag. (C) HEK293 cells were co-transfected with Ubiquitin-HA, DDX6-myc, and control siRNA or siRNA targeting endogenous praja2. Cells were starved 24 h, pre-treated with MG132 (10 μM), then treated with Forskolin (Fsk) (20 μM) for 3 h and immunoprecipitated with anti-myc. Lysates were immunoblotted with anti-praja2, while precipitates with anti-HA (DDX6 poly-Ub) and anti-myc. (D) HEK293 cells were co-transfected with Ubiquitin-HA, DDX6-myc, and empty vector (NT) or flag-praja2dm. Cells were starved 24 h, pre-treated with MG132 (10 μM), then treated with Forskolin (Fsk) (20 μM) for 3 h and immunoprecipitated with anti-myc. Lysates were immunoblotted with anti-flag, while precipitates with anti-HA (DDX6 poly-Ub) and anti-myc. (E) Schematic model showing the lysine residues of DDX6 mutated in arginine in the allK/R mutant. (F) HEK293 cells were co-transfected with Ubiquitin-HA and DDX6-myc (wild-type, K$_{26,94,118,286}$R or allK/R). Cells were starved 24 h, pre-treated with MG132 (10 μM), then treated with Fsk (20 μM) for 3 h and immunoprecipitated with anti-myc. Precipitates were immunoblotted with anti-HA (DDX6 poly-Ub) and anti-myc. (G) HEK293 cells were co-transfected with Ubiquitin-HA and DDX6-myc. Cells were starved 24 h, pre-treated with MG132 (10 μM), then treated with Fsk (20 μM) for 3 h and immunoprecipitated with anti-myc. Precipitates were immunoblotted with anti-Ub-K63 and anti-myc. (H) Same as in (F). Precipitates were immunoblotted with anti-Ub-K48 and anti-myc. (I) Schematic picture showing that cAMP induces praja2-mediated DDX6 ubiquitylation. Source data are available online for this figure.

transferred by praja2 by analyzing the available databases reporting ubiquitylated lysines in this RNA helicase (Hornbeck et al, 2015, 2019). A guided site-directed mutagenesis and ubiquitylation assays revealed that none of the previously reported ubiquitylated lysines was, indeed, targeted by praja2 (Appendix Fig. S1). To survey potential lysine residues of DDX6 accepting ubiquitin molecules in response to cAMP stimulation, we used an unbiased approach. Specifically, we affinity-purified streptavidin (STREP)-tagged DDX6 from HEK293T cells treated with Fsk or left untreated. After purification, the purified DDX6 was analyzed by liquid chromatography-tandem mass spectrometry (LC-MS/MS) to identify di-glycine tag (GG) on ubiquitin-modified lysine residues after tryptic digestion. Our analysis showed that lysine 26 and 286 were GG-modified upon Fsk treatment, thus indicating that these 2 residues are specifically ubiquitylated upon cAMP stimulation (Appendix Fig. S2A–C). Importantly, the difference in the enrichment of these two modified peptides in the Fsk-treated group was not due to the difference in the amount of purified DDX6 as we obtained virtually identical amount of bait in the groups DDX6 and DDX6+Fsk (Appendix Fig. S2D). In addition, our results also indicated that the GG-modification of lysine 118 occurs predominantly in non-treated cells, thus suggesting that the ubiquitylation profile of DDX6 may be differentially shaped in response to different cellular cues (Appendix Fig. S2A). Anyway, the mutagenesis of the identified lysin residues was not sufficient to abrogate DDX6 ubiquitylation in cAMP-stimulated conditions (Fig. 4E,F). In fact, site-directed mutagenesis and ubiquitylation assays demonstrated that multiple lysine residues contribute to DDX6 ubiquitylation in cAMP-stimulated cells. In particular, substitution of lysine 26, 67, 86, 94, 104, 118, 146, 195, 286, 458, and 482 to arginine (DDX6 allK/R) mostly abrogated DDX6 polyubiquitylation in Fsk-treated cells (Fig. 4E,F). The ubiquitylation of DDX6 was non-proteolytic, since we observed mostly lysine 63-linked polyubiquitylation with no significant changes of DDX6 levels upon cAMP stimulation (Fig. 4G,H). These results demonstrated that cAMP induces praja2-mediated DDX6 ubiquitylation (Fig. 4I).

## cAMP–praja2 axis controls P-body dynamics

P-bodies are non-membranous cytoplasmic condensates of ribonucleoprotein complexes that control the translation/decay of mRNAs, thus playing a fundamental role in post-transcriptional control of cell functions (Parker and Sheth, 2007). Signaling pathways have been identified as important regulators of P-bodies (Wei et al, 2022; Bearss et al, 2021; Vidal-Cruchez et al, 2023). Yet, the role of GPCR-cAMP signaling in P-body assembly and translation repression is still unknown. To fill this gap, we explored the effect of cAMP activation on P-body assembly. Human GBM cells (U87MG) were treated with Fsk, fixed at different time points from stimulation and subjected to double immunostaining for DDX6 and DCP1A. As shown in Fig. 5A–C, Fsk treatment induced a significant increase of P-body number. A similar finding was obtained in two primary cultures of GBM cells derived from biopsies of human GBM tissues (Figs. 5D–F and EV4A–C). Virtually identical results were obtained by treating cells with prostaglandin E2 (PGE2), a ligand of the G-protein-coupled receptor EP2 and activator of the cAMP pathway (Bidwell et al, 2010) (Fig. 5G,H). Pre-treatment with the PKA inhibitor KT5720 completely reversed the effects of PGE2 on P-bodies (Fig. 5G,H). Next, we evaluated the effects of wild-type praja2 or praja2dm on P-body formation in U87MG cell line. In non-stimulated cells, overexpression of praja2 was sufficient to increase the number of P-bodies, compared to control cells or to cells expressing praja2dm mutant. Treatment with Fsk induced a significant increase of P-bodies number in control cells that was inhibited by over-expressing praja2dm (Fig. 5I,J). To confirm that the effects of cAMP stimulation on P-body condensation were related to praja2 activity, we generated a praja2-knockout (praja2KO) U87MG cell line by CRISPR/Cas9 strategy. Figure 6A–C shows that activation of cAMP stimulated the assembly of P-bodies in wild-type cells (WT), whereas the number of P-bodies was markedly affected in praja2KO cells. This effect was confirmed also treating cells with PGE2 (Fig. 6D–F). Rescue experiments with praja2 restored the cAMP-induced assembly of P-bodies in praja2KO cells (Fig. 6A–C). Collectively, these results demonstrate that activation of GPCR-cAMP-PKA-praja2 axis promotes the assembly of P-bodies (Fig. 6G).

## Praja2 regulates the assembly of DDX6/mRNA complexes and translation repression in human glioblastoma

P-body constitutes an important hub for the translational regulation of mRNAs (Brengues et al, 2005). Considering the role played by praja2 in the regulation of P-body dynamics, we

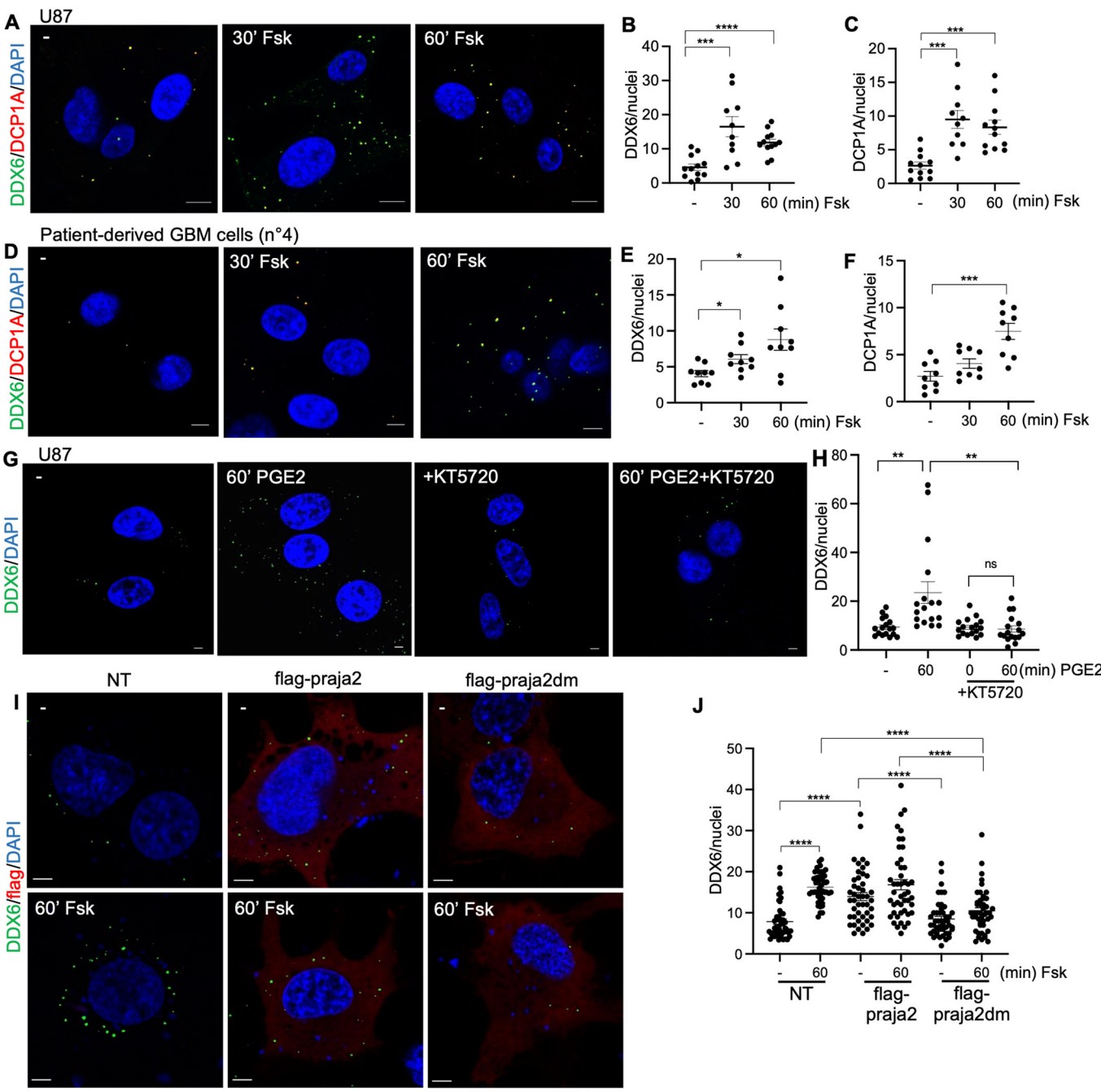

**Figure 5. cAMP promotes the assembly of P-bodies.**

(A) U87MG cells were treated 30 and 60 min with Forskolin (Fsk) (10 μM). Cells were fixed and immunostained with anti-DDX6, anti-DCP1A, and DAPI. Scale bar: 10 μm. (B, C) Quantitative analysis of four biological independent experiments, mean ± SEM is indicated. *t* test ***P < 0.001, ****P < 0.0001 (B, 30 Fsk = 0.0005) (C, 30 Fsk = 0.0005; 60 Fsk = 0.0002). (D) GBM cells derived from patient 4 were treated 30 and 60 min with Forskolin (10 μM). Cells were fixed and immunostained with anti-DDX6, anti-DCP1A, and DAPI. Scale bar: 5 μm. (E, F) Quantitative analysis of three biological independent experiments, mean ± SEM is indicated. *t* test *P < 0.05, ***P < 0.001 (E, 30 Fsk = 0.0184; 60 Fsk = 0.0135) (F, 60 Fsk = 0.0003). (G) U87MG cells were treated with prostaglandin E2 (PGE2) (1 μM) for 1 h and where indicated, pre-treated 30 min with KT5720 (10 μM). Cells were fixed and stained with anti-DDX6 and DAPI. Scale bar: 5 μm. (H) Statistical analysis of three biological independent experiments, mean ± SEM is indicated. *t* test **P < 0.01 (− vs 60 PGE2 = 0.0039; 60 PGE2 vs 60 PGE2 + KT5720 = 0.003). (I) U87MG cells overexpressing an empty vector (NT), flag-praja2 or flag-praja2dm mutant were treated for 60 min with Fsk (10 μM), fixed and stained with anti-DDX6 and anti-flag antibodies, and DAPI. Scale bar: 5 μm. (J) Statistical analysis of four biological independent experiments, mean ± SEM is indicated. *t* test ****P < 0.0001. Source data are available online for this figure.

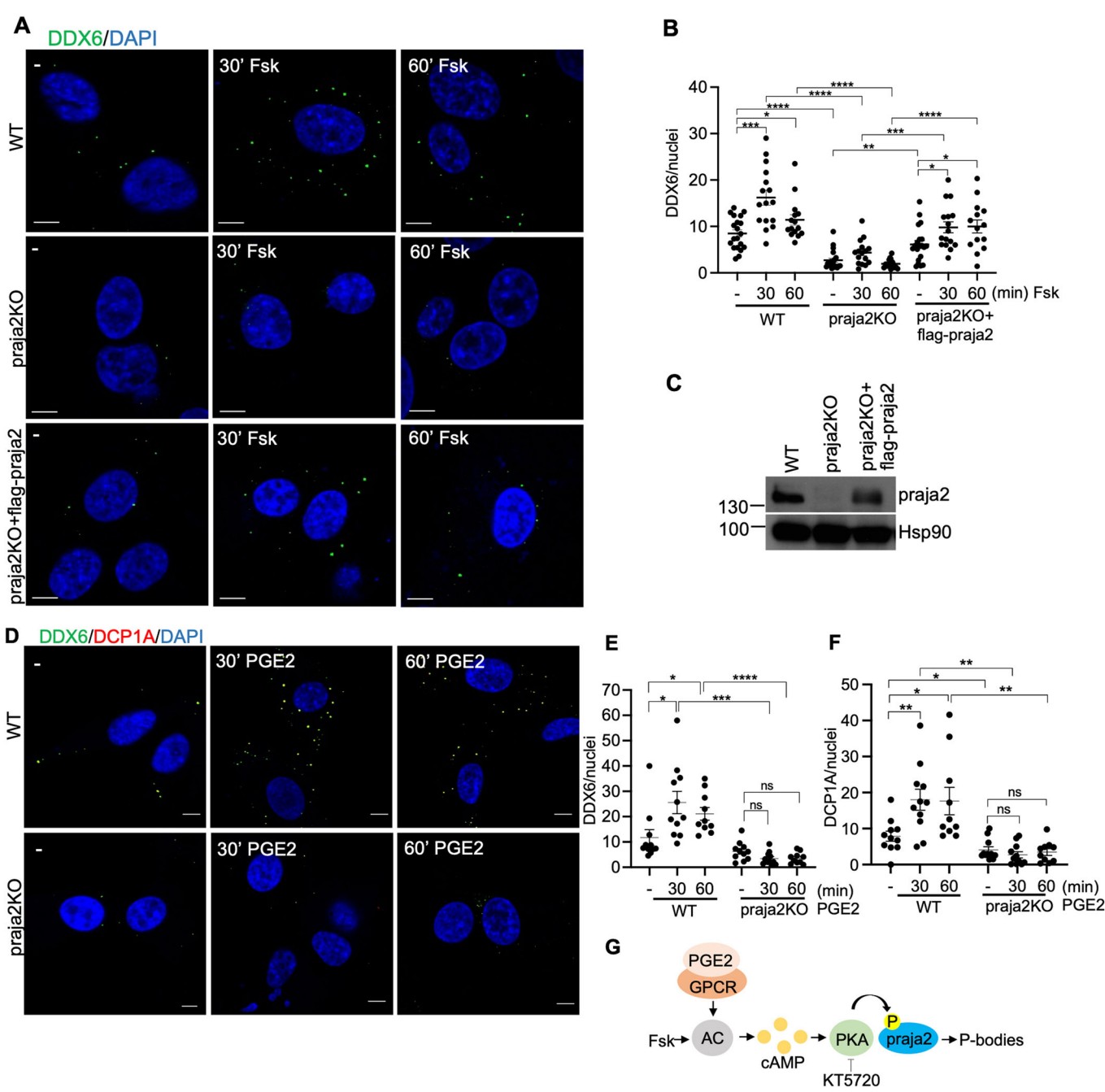

**Figure 6. Praja2 controls P-body dynamics.**

(A) U87MG (WT), U87MG praja2KO, and U87MG praja2KO cells stably expressing flag-praja2 were treated 30 and 60 min with Fsk (10 μM). Cells were fixed and immunostained with anti-DDX6 and DAPI. Scale bar: 5 μm. (B) Quantitative analysis of five biological independent experiments, mean ± SEM is indicated. t test *P < 0.05, **P < 0.01, ***P < 0.001, ****P < 0.0001 (WT− vs WT 30 Fsk=0.0003; WT− vs WT 60 Fsk= 0.0421; praja2KO− vs praja2KO+flag-praja2− = 0.0014; praja2KO 30 Fsk vs praja2KO+praja2-flag 30 Fsk= 0.0006; praja2KO+flag-praja2− vs praja2KO+flag-praja2 30 Fsk = 0.0175; praja2KO+flag-praja2− vs praja2KO+flag-praja2 60 Fsk= 0.0265). (C) Lysates from U87MG (WT), U87MG praja2KO and U87MG praja2KO+flag-praja2 cells were immunoblotted with anti-praja2 and anti-Hsp90. (D) U87MG (WT) and U87MG praja2KO cells were treated for 30–60 min with Prostaglandin E2 (PGE2) (1 μM). Cells were fixed and immunostained with anti-DDX6, anti-DCP1A, and DAPI. Scale bar: 5 μm. (E, F) Quantitative analysis of four biological independent experiments, mean ± SEM is indicated. t test *P < 0.05, **P < 0.01, ***P < 0.001, ****P < 0.0001 (E, WT - vs WT 30 PGE2 = 0.0189; WT− vs WT 60 PGE2 = 0.0299; WT 30 PGE2 vs prajaKO 30 PGE2 = 0.0005) (F, WT− vs WT 30 PGE2 = 0.0072; WT− vs WT 60 PGE2 = 0.034; WT− vs praja2KO 0 = 0.034; WT 30 PGE2 vs praja2KO 30 PGE2 = 0.0003; WT 60 PGE2 vs praja2KO 60 PGE2 = 0.0047). (G) Schematic picture showing that cAMP induces P-body assembly. Source data are available online for this figure.

evaluated the impact of praja2 downregulation on mRNA translation rate by performing polysome profiling (PP) analysis in GBM cells. Analysis of the area under the curve showed that the ratio between actively translating polysomes and the sub-polysomal fractions was higher in praja2KO GBM cells compared to wild-type cells, suggesting that praja2 represses the global mRNAs translation (Fig. 7A,B; Appendix Fig. S3A). To identify transcripts translationally regulated by praja2, we performed a PP-sequencing in wild-type and praja2KO GBM cells. We applied poly-A RNAseq to both purified polysomes (fractions 6–11) and input fractions, and then analyzed our data with the Riborex algorithm. The analysis identified 515 mRNAs that differentially associate (log2FC > 0.5 and Padj ≤0.05) with actively translating polysomes in the absence of praja2 (Fig. 7C; Dataset EV4). To test if the different association with the polysomal fraction was reflecting a different amount of protein, we analyzed protein levels of selected mRNAs (TF and CTNNA3) whose polysome association was increased in praja2KO cells. The immunoblot data confirmed the results of PP-sequencing analysis showing a significant increase of TF and CTNNA3 proteins in praja2KO cells, compared to controls (Fig. 7D–G).

According to the regulation of P-body assembly by praja2, we evaluated potential changes in DDX6/mRNA complexes in wild-type and in praja2KO GBM cells by DDX6 RNA-immunoprecipitation (RIP) and sequencing analysis. First, we compared the transcriptional profile of wild-type and praja2KO GBM cells. We identified transcripts that were differentially expressed between these two cell types, as illustrated in the volcano plot (Fig. EV5A; Dataset EV5). In particular, we found 960 downregulated transcripts (in green) and 544 upregulated transcripts (in red) in praja2KO compared to wild-type GBM cells (Fig. EV5A), confirming that praja2 regulates the transcriptome profile of GBM cells. An ingenuity pathway analysis (IPA) revealed the involvement of differentially expressed mRNAs in important biological pathways, such as cellular stress, cell growth, development, intracellular signaling cascades, and cancer (Fig. EV5B; Dataset EV6). Next, we analyzed the mRNAs differentially bound to DDX6 in wild-type and praja2KO GBM cells. Immunoprecipitation of DDX6 was specific since no signal was detected in control samples (IgG) (Appendix Fig. S3B). As shown in Fig. 7H, the number of DDX6-bound mRNAs was increased in praja2KO cells, compared to wild-type cells. Moreover, although the vast majority of DDX6 targets were consistently identified in both conditions, we found that specific groups of mRNAs were complexed with DDX6 exclusively in wild-type or in praja2KO cells, indicating that praja2 influences the binding of DDX6 to selected mRNAs (Fig. 7H; Dataset EV7). We validated this finding by performing independent RIP-qPCR analyses, which confirmed our sequencing data (Fig. 7I). Then, we asked whether the regulation of mRNAs translation mediated by praja2 was dependent, at least in part, on the impact of the E3 ligase on P-body dynamics and DDX6/mRNAs complexes. To test this hypothesis, we compared the sequencing data from polysome profiling with those obtained with DDX6-RIP analysis. We were particularly interested in comparing mRNAs translationally regulated by praja2 with those bound by DDX6 only in praja2KO cells. This overlap led to the identification of 23 common mRNAs (Fig. 7J; Table EV1). A gene ontology (GO) analysis revealed that these transcripts are involved in different biological processes functionally linked to GBM (Fig. 7K, left panel; Dataset EV8). As example, the positive regulation of axonogenesis pathway

is associated to the aggressive nature of the tumor and to glioma cells motility (Sun et al, 2024; Shao et al, 2022). Similarly, we found a negative regulation of vascular permeability pathway that underlies angiogenesis and GBM cell proliferation (Das and Marsden, 2013; Treps et al, 2016). A fundamental role in GBM progression is played by tumor-associated macrophages that promote angiogenesis and proliferation (Qian and Pollard, 2010) and inhibitors of macrophage migration have been used as anti-angiogenic therapy in GBM (Castro et al, 2017; Mangano et al, 2018). Interestingly, among the differentially expressed transcripts, we identified inhibitors of macrophages and leukocyte migration. Similarly, a pathway enrichment analysis demonstrated the involvement of these transcripts in pathways related to GBM, such as the interferon regulatory transcription factor 3 (IRF3)-Interferon (IFN) cascade, components of the innate immunity system (Pattwell and Holland, 2017), ROBO receptors involved in cell migration and invasion (Markouli et al, 2024; Geraldo et al, 2021) and β-oxidation of fatty acids that is critical for the metabolic adaptation of GBM cells (Miska and Chandel, 2023) and a valuable therapeutic target (Lin et al, 2017) (Fig. 7K, right panel; Dataset EV9). Taken together, these results indicate that praja2 regulates mRNA translation in GBM cells and this mechanism depends, at least in part, on praja2-mediated regulation of DDX6/mRNAs complexes and P-body dynamics.

## Deletion of praja2 induces premature senescence and GBM growth arrest

Cellular senescence is a physiological process that controls different aspects of tissue remodeling, repair following wound healing and embryonic development. Prolonged senescence underlies several pathological conditions, including cancer development, degenerative disorders and aging-associated diseases (Kritsilis et al, 2018; Martínez-Cué and Rueda, 2020; Schmitt et al, 2022). The principal mechanism of cellular senescence is the arrest of cell cycle at G1 or G2 phase and ultimately inhibition of cell growth (Kumari and Jat, 2021). Previous work established a central role of praja2 in GBM growth (Lignitto et al, 2013; Delle Donne et al, 2022). Accordingly, we assessed whether praja2KO GBM cells activate the program of cellular senescence to inhibit proliferation. First, we evaluated the activity of lysosomal senescence-associated β-galactosidase enzyme (Lee et al, 2006), observing a marked increase of β-gal staining in praja2KO cells, compared to wild-type counterpart (Fig. 8A,B). Next, we analyzed the expression levels of p21$^{CIP1}$, a well-known gene product that accumulates in senescent cells to induce growth arrest (Mansour et al, 2023). Consistently, we found a significant increase of p21 protein levels in praja2KO cells, compared to controls (Fig. 8C,D), indicating that downregulation of praja2 induces a program of premature cellular senescence. Since cellular senescence is often coupled to growth arrest, we evaluated the effects of praja2 downregulation on tumor growth in vivo. Wild-type or praja2KO GBM cells were subcutaneously injected into CD1 nude mice and tumor growth was evaluated overtime. Six weeks later, mice were sacrificed, and tumor lesions were isolated and further analyzed. Figure 8E, F shows that tumor volume was markedly reduced in mice injected with praja2KO cells compared to controls. Immunohistochemistry (IHC) confirmed a marked reduction of the proliferative marker Ki67 in praja2KO tumor samples, compared to wild-type ones (Fig. 8G; Appendix Fig. S4A).

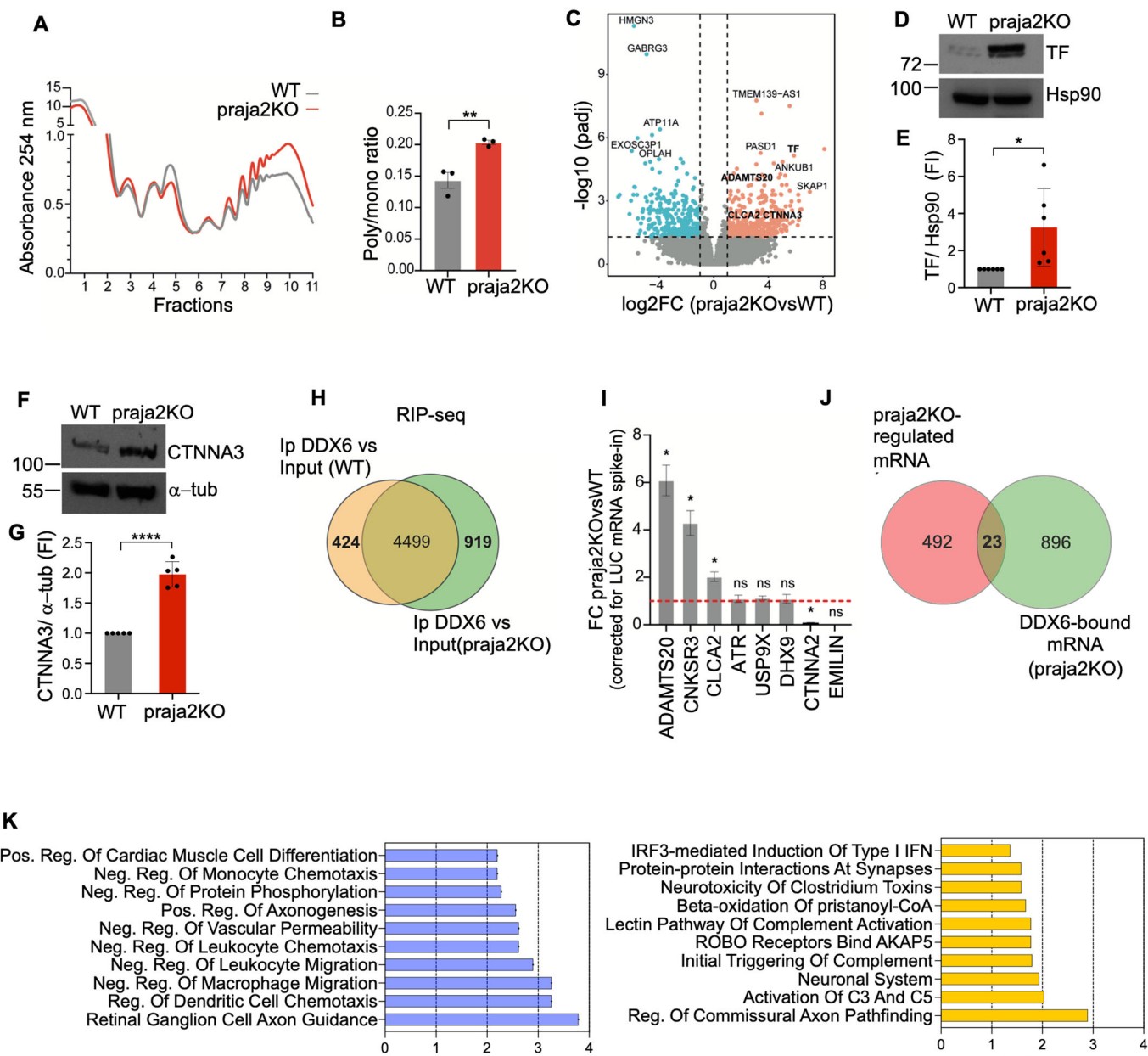

**Figure 7. Praja2 regulates DDX6-mRNA complexes and mRNA translation.**

(A) Typical polysome profiles of wild-type (WT) and praja2KO U87MG cells. (B) Quantification of the polysome/monosome area ratio in wild-type and praja2KO profiles as reported in (A) (n = 3 biological independent replicates). Error bars represent SEM. Unpaired t test **P = 0.0096. (C) Volcano plot showing the significant differentially polysome-bound mRNAs in praja2KO vs U87MG wild-type cells (n = 3 biological independent replicates). The statistical analysis was performed with Wald test of Deseq2. (D) Lysates from U87MG WT and praja2KO cells were immunoblotted with anti-TF and anti-Hsp90. (E) Quantitative analysis of six biological independent experiments as reported in (D), mean ± SD is indicated. t test *P = 0.0254. (F) Lysates from U87MG WT and praja2KO cells were immunoblotted with anti-CTNNA3 and anti-α-tubulin. (G) Quantitative analysis of five biological independent experiments as reported in (F), mean ± SD is indicated. t test ****P < 0.0001. (H) Overlap between DDX6-bound mRNAs in WT and praja2KO U87MG cells. (I) Validation of DDX6 targets by RT-qPCR. Results are shown as fold change of praja2KO vs WT U87MG cells (n = 3). Levels of immunoprecipitated mRNA were corrected for Luciferase mRNA spike-in control. EMILIN was used as a negative control. Error bars represent SEM. Statistical significance was assessed by one-sample t test *P < 0.05 (ADAMTS20 = 0.016; CNKSR3 = 0.0243; CLCA2 = 0.0371; CTNNA2 < 0.0001). (J) Overlap between praja2 translationally regulated mRNAs and DDX6 mRNAs bound only in the absence of praja2. (K) Gene ontology analysis of 23 transcripts identified in (J). Biological processes (left panel) and pathway enrichment analysis (right panel) are shown. Source data are available online for this figure.

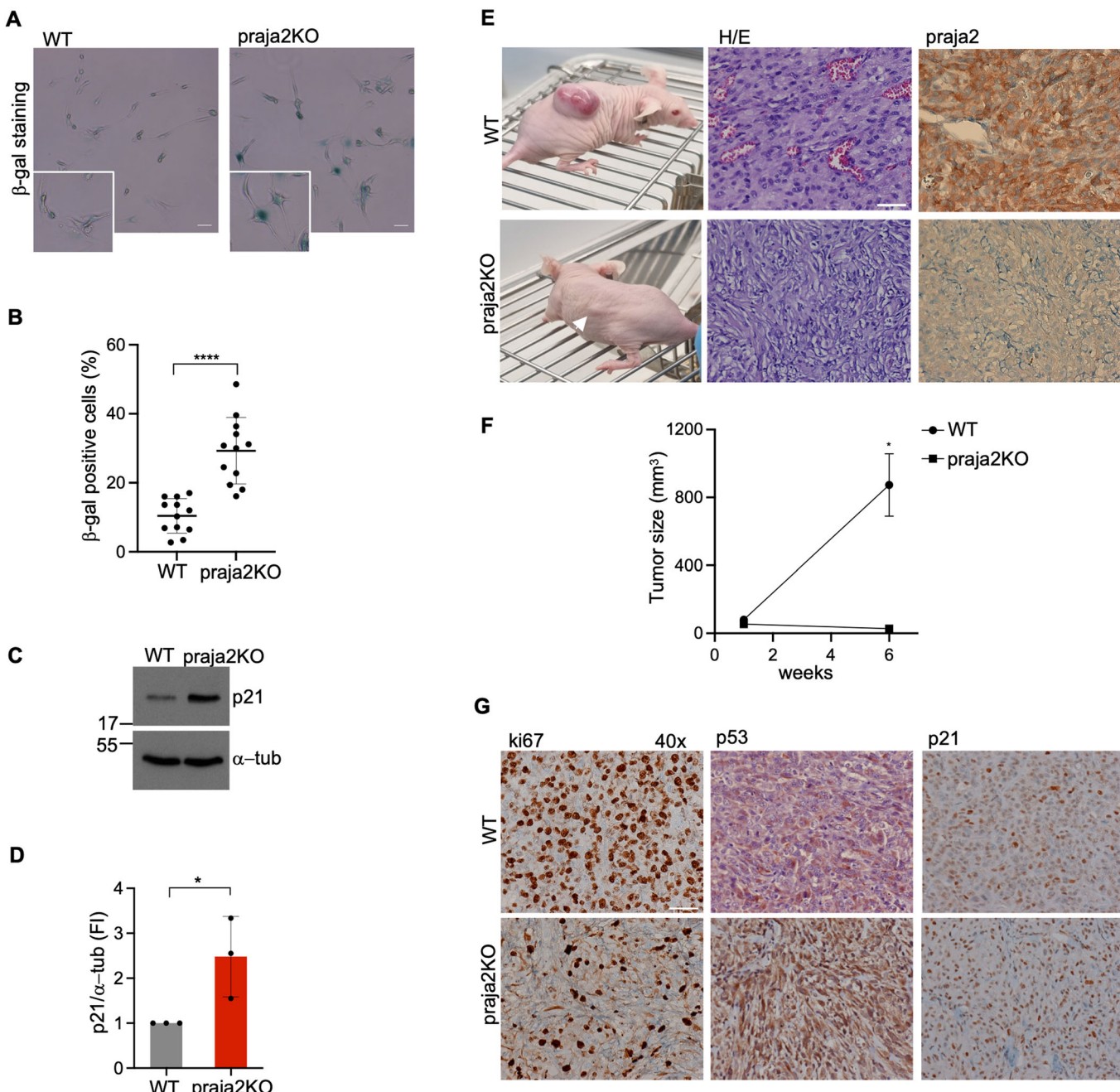

**Figure 8. Praja2 deletion induces cellular senescence and GBM growth arrest.**

(A) β-galactosidase staining at pH 6 on U87MG WT and praja2KO cells. Scale bar: 40 μm. (B) Quantitative analysis of three biological independent experiments as reported in (A), mean ± SD is indicated. *t* test ****$P < 0.0001$. (C) Lysates from U87MG (WT) and praja2KO cells were immunoblotted with anti-p21 and anti-α-tubulin. (D) Quantitative analysis of three biological independent experiments as reported in (C), mean ± SD is indicated. *t* test *$P = 0.0459$ (E). $5 \times 10^6$ U87MG (WT) and U87MG praja2KO cells were implanted into CD1 nude mice. Six weeks later, the mice were killed, and the tumors were excised and weighed. Tumor sections were fixed and doubly stained with hematoxylin/eosin or subjected to immunohistochemistry with anti-praja2 antibody. Scale bar (referred to ×40 magnification): 50 μm. (F) Quantitative analysis of the tumors size (at 6 weeks) shown in (E). Three WT and four praja2KO mice were analyzed. Data are expressed as mean value ± SEM. *t* test *$P = 0.0441$ (G). Representative imaging of immunostaining for ki67, p53, and p21 in mice subcutaneously injected with U87MG (WT) (upper panels) or U87MG praja2KO (lower panels) cells. Scale bar (referred to ×40 magnification): 50 μm. Source data are available online for this figure.

Conversely, the immunostaining of p53 and p21 was upregulated in praja2KO tumor sections, compared to controls (Fig. 8G; Appendix Fig. S4B,C). These findings confirmed the essential role of praja2 in GBM growth in vivo and demonstrated that its downregulation activates the p53/p21 $^{CIP 1}$-mediated senescence pathway.

## DDX6 ubiquitylation controls P-body dynamics and GBM growth

We demonstrated that praja2 regulates P-body dynamics, mRNA translation, and GBM growth. To understand whether this regulation depends, at least in part, on praja2-mediated DDX6 ubiquitylation, we evaluated the effects of DDX6 allK/R mutant on P-body assembly. As shown in Fig. 9A–C, Fsk treatment induced an increase in the number of P-bodies, both in control cells and in cells overexpressing wild-type DDX6. In contrast, overexpression of DDX6 allK/R mutant markedly impaired condensation of P-bodies in Fsk-treated cells (Fig. 9A–C), indicating that K63-linked ubiquitylation of DDX6 by praja2 in response to cAMP stimulation is required for P-body assembly. This finding is consistent with the emerging role of K63 ubiquitylation in the regulation of protein liquid-liquid phase separation (Dao et al, 2022). The role of P-bodies in tumor growth has been described (Nsengimana et al, 2022). Accordingly, we evaluated if praja2-mediated DDX6 ubiquitylation, by regulating P-body assembly, impacts on tumor growth. To this end, we subcutaneously injected U87MG cells stably expressing DDX6 wild-type, DDX6 allK/R mutant or empty vector into nude mice and measured tumor growth over a time-point curve. As shown in Fig. 9D–G, expression of the DDX6 ubiquitylation mutant strongly accelerated tumor growth, compared to wild-type DDX6 or empty vector. These findings indicate that cAMP–praja2–DDX6 axis regulates dynamic of P-bodies and GBM growth.

## Discussion

Here, we report the identification of an integrated ubiquitin network operating in glioblastoma cells that controls mRNA translation. Ubiquitylation of the RNA helicase DDX6 by the RING ligase praja2 in response to cAMP signaling controls the assembly of DDX6/mRNA complexes and P-body formation, with important implications for the translational regulation of key target mRNAs and GBM growth (Fig. 9H).

The control of mRNA fate is a fundamental homeostatic mechanism that regulates gene expression underlying cell growth and survival, differentiation, and development. Once exported from the nucleus to the cytoplasm, mature mRNAs are complexed with dedicated RNA-binding proteins (RNPs) and can be translated, stored or degraded (Thelen and Kye, 2019; Dreyfuss et al, 2002). The translation rate depends, at least in part, by a dynamic balance between polysome-bound mRNAs and mRNAs-RNPs condensates at P-bodies (Parker and Sheth, 2007). In these organelles, ribonucleoprotein complexes repress the translation of mRNAs or promote their degradation, thus constituting an essential post-transcriptional control mechanism to dynamically regulate gene expression.

Here, we added a novel layer of complexity to the biological processes of mRNA translation by identifying a regulatory mechanism operated by the GPCR-cAMP cascade that, in response to hormone stimulation, promotes the assembly of P-bodies. We found that the E3 ubiquitin ligase praja2 localizes at P-bodies and forms a complex with core components of these organelles, including RNA helicase DDX6, DCP1A, and EDC3. Binding assays, site-directed mutagenesis, and genetic analysis in yeast confirmed that praja2 and DDX6 directly interact through dedicated binding domains. Praja2 acts in response to cAMP stimulation to control essential aspects of cell physiology (Lignitto et al, 2011; Senatore et al, 2021; Chiuso et al, 2023). In GBM cells, upregulation of praja2 stimulates the mitogenic cascade, inhibits oncosuppressive pathways and promotes the metabolic rewiring underlying tumor growth (Lignitto et al, 2013; Delle Donne et al, 2022). Single cell transcriptomic analysis in GBM tissues revealed a significant correlation between the expression profile of praja2 and specific mRNAs involved in different aspects of cancer cell behavior. The in vivo data were supported by genetically manipulating praja2 levels in GBM cells. Moreover, proteomic analysis of affinity-purified praja2 complexes confirmed a physical interaction between the ligase and components of P-bodies, pointing to the role of the ligase in the control of translation in GBM.

Different signaling pathways and post-translational events have been identified as relevant players in the assembly/disassembly of P-bodies (Tenekeci et al, 2016; Yoon et al, 2010; Chiang et al, 2013; Arribas-Layton et al, 2016; Zhan et al, 2024). However, the relevance of GPCR-mediated cAMP signaling and the mechanism controlling P-body dynamics in response to hormone stimulation was largely unexplored. Our data demonstrate that praja2, in response to cAMP stimulation, ubiquitylates DDX6 at multiple lysine residues. Ubiquitylation does not lead to proteolysis of DDX6, instead it controls its targeting to P-bodies. The finding that praja2 staining only colocalizes with some P-bodies suggests that praja2 binding to DDX6 and its localization at P-bodies is a dynamic process possibly regulated by signaling events in response to specific cellular needs or extracellular stimuli. The mechanism operated by praja2 may also involve other components of P-bodies, including EDC3 and DCP1A, with important implications for organelle biology and dynamics. Nevertheless, the findings that deletion of praja2 or expression of a DDX6 ubiquitylation-defective mutant decreases the number of P-bodies indicate that PKA-praja2-DDX6 axis is, indeed, a relevant mechanism for the organelle assembly.

The non-proteolytic control of protein fate by praja2 has been previously reported (Zhong et al, 2017; Chiuso et al, 2023; Rinaldi et al, 2024; Schiefer and Hale, 2024). The effects of cAMP on the organelles were abrogated by targeted deletion of praja2, indicating the existence of a mechanism of P-body condensation operated by the ubiquitin system. Consistently, expression of an ubiquitylation-defective DDX6 mutant prevented assembly of P-bodies. Regulatory ubiquitylation of other resident proteins of P-bodies, including DCP1A, EDC4 and 4-ET, have been implicated in the organelles condensation (Tenekeci et al, 2016; Wei et al, 2022; Kedia et al, 2022). The mechanism by which ubiquitylation regulates dynamic of P-bodies is not known. An emerging concept suggests that ubiquitylation defines a dynamic and flexible three-dimensional code that regulates the assembly/disassembly of multimeric protein complexes at distinct intracellular compartments (Chiuso et al, 2023). The RNAseq data on immunopurified DDX6 also suggests

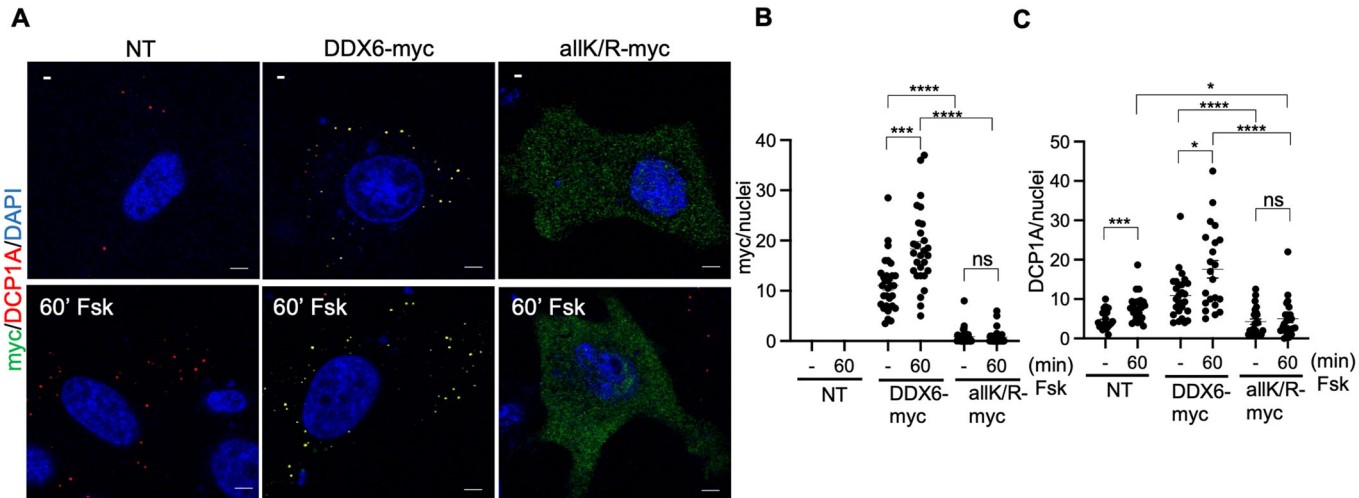

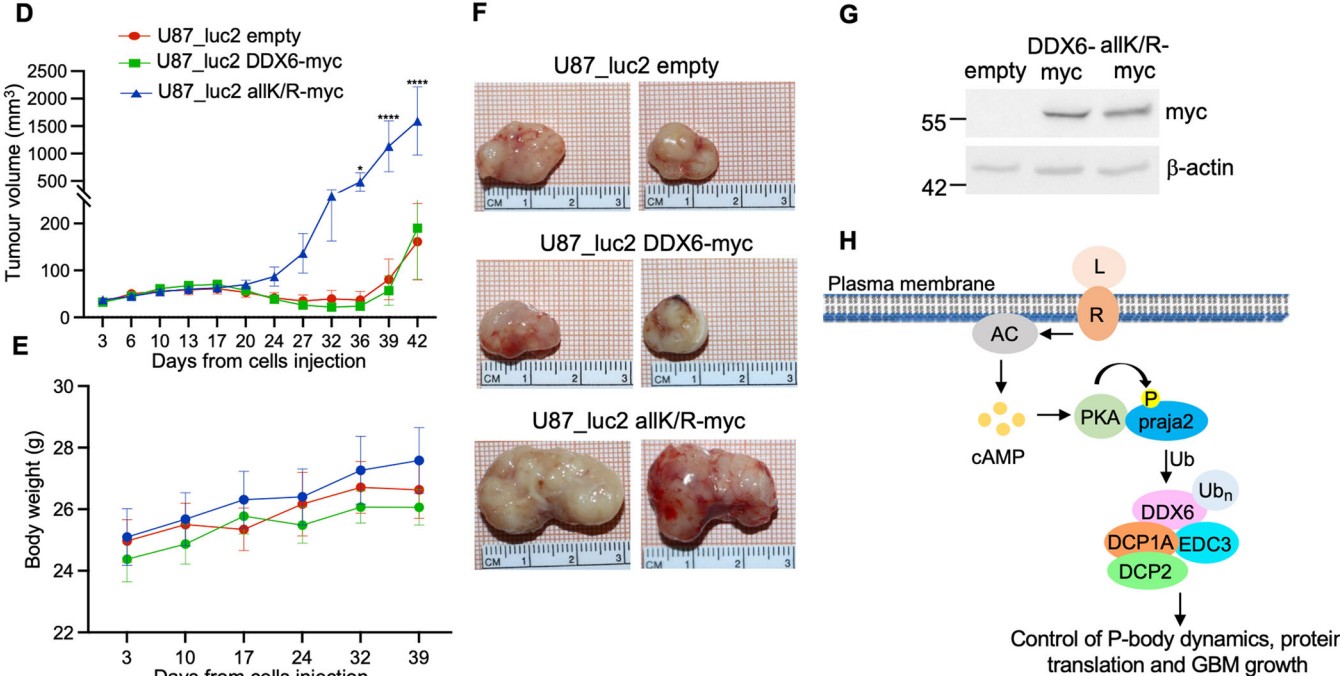

**Figure 9. DDX6 ubiquitylation controls P-bodies assembly and GBM growth.**

(A) U87MG cells overexpressing an empty vector (NT), DDX6-myc or DDX6 allK/R-myc mutant were treated for 60 min with Fsk (10 μM), fixed and immunostained with anti-DCP1A, anti-myc and DAPI. Scale bar: 5 μm. (B, C) Quantitative analysis of five biological independent experiments is shown, mean ± SEM is indicated. *t* test *P < 0.05, ***P < 0.001, ****P < 0.0001 (B, DDX6-myc− vs DDX6-myc 60 Fsk= 0.0001) (C, NT – vs NT 60 Fsk = 0.0003; DDX6-myc− vs DDX6-myc 60 Fsk = 0.0115; NT 60 Fsk vs allK/R-myc 60 Fsk= 0.0251). (D) In all, 2.5 × 10^6 U87MG-luc cells infected with empty vector, or with a vector expressing DDX6-myc or allK/R-myc mutant were subcutaneously implanted into nude mice. Cumulative data of tumor volume measured twice a week are expressed as mean ± SEM. Six mice for each experimental group were used. two-way ANOVA *P < 0.05, ****P < 0,0001 (36 days empty vs allK/R-myc = 0.0231; 36 days DDX6-myc vs allK/R-myc = 0.0186). (E) Cumulative data of mice body weight measured once a week are expressed as mean ± SEM. Six mice for each experimental group were used. (F) Representative images of tumors excised 46 days post-injection are shown. (G) Lysates from U87MG_luc cells infected with empty virus or with lentiviral particles carrying DDX6-myc or allK/R-myc transgene were immunoblotted with anti-myc and anti-β-actin antibodies. (H) Stimulation of G-protein-coupled receptor (R) by a ligand (L) activates adenylate cyclase (AC), raises cAMP levels, and activates protein kinase A (PKA). PKA phosphorylates and activates praja2 that, in turns, polyubiquitylates DDX6. Ubiquitylated DDX6 controls P-body assembly, mRNA translation and GBM growth. Source data are available online for this figure.

that ubiquitylation of the helicase by praja2 could be involved in mediating protein/mRNAs complex formation. Therefore, the ubiquitin system may act at different layers of complexity by regulating protein–protein interactions and/or formation of protein/RNA complexes, with important implications for cancer cell biology. An important aspect that needs to be addressed in a next future is the identification of relevant deubiquitinases, also known as ubiquitin hydrolases, which may remove ubiquitin moieties from ubiquitylated proteins, dynamically reversing the effects of the ubiquitin pathway on P-body condensation and mRNA fate.

The regulation of P-body assembly by the cAMP-PKA pathway represents an important mechanism to control protein translation. In yeast, PKA restrains P-bodies formation through direct phosphorylation of Pat1 and consequent inhibition of Dhh1 (protein ortholog of human DDX6) recruitment at nucleating organelles. The inactivation of PKA is, indeed, sufficient for assembly of P-bodies (Ramachandran et al, 2011). However, the role of PKA in the control of P-body dynamics, DDX6-mRNA interaction and protein translation in mammalian cells was largely unknown so far. Here, we have originally demonstrated that rise of cAMP levels induces assembly of P-bodies in human GBM cells. This finding agrees with recent evidence indicating that stimulation of cAMP signaling induces a liquid-liquid phase separation (LLPS) of key regulatory components of this signaling cascade. In particular, the regulatory subunit of PKA RIα, upon GPCR stimulation, forms biomolecular condensates that retain high cAMP levels and PKA activity (Zhang et al, 2020). The control of P-body dynamics and translation repression by GPCR ligands may, thus, represent a phylogenetically conserved mechanism of cellular fitness that supports cell growth and survival under stress conditions or changed extracellular microenvironment. This aspect is of particular importance for cancer cells that undergo rapid metabolic changes in response to modified tumor microenvironment or in the course of chemotherapy. In this context, the role of P-bodies in tumor growth and cancer dissemination has been described (Bearss et al, 2021; Hardy et al, 2017; Shen et al, 2024; Yoshino et al, 2020; Chen et al, 2022; Fang et al, 2024; Zhan et al, 2024). Nevertheless, the impact of P-bodies in GBM growth was not previously studied. In GBM tissues, praja2 is overexpressed and its overrepresentation markedly promotes tumor growth (Lignitto et al, 2013; Delle Donne et al, 2022). Here, we have added a novel layer of complexity to the dynamic interface between hormone signaling and GBM growth, demonstrating the existence of a cAMP–praja2-DDX6 signaling axis that finely controls P-body assembly and mRNA translation. We found that genetic deletion of praja2 significantly increased the global protein translation in tumor cells. This is an important mechanism adopted by cancer cells to rapidly adapt to external stimuli, contributing to the deleterious plasticity of malignant tumors (Bhat et al, 2015; Robichaud et al, 2019; Lee et al, 2021; Galardi et al, 2016; Wahba et al, 2016). In GBM cells, the activation of Ras-AKT oncogenic signaling regulates the physical and functional association of mRNAs to active polysomes, supporting anabolic pathways (Montiel-Dávalos et al, 2023; Rajasekhar et al, 2003). The identification of praja2 as a regulator of global protein translation contributes as a relevant mechanism for GBM growth and dissemination. We have previously reported that praja2 promotes

cancer growth by regulating different signaling cascades, like Hippo and MAPK (Lignitto et al, 2013; Rinaldi et al, 2016; Zhao et al, 2021). Indeed, targeting praja2 is a valid strategy to arrest GBM growth (Delle Donne et al, 2022). Accordingly, we report that praja2KO GBM cells undergo premature senescence in vitro and failed to develop tumors in vivo, in agreement to the established role of cellular senescence as a growth-arrested state and an anti-tumor mechanism (Hornsby, 2007; Pérez-Mancera et al, 2014). Interestingly, expression of DDX6 ubiquitylation mutant promotes P-body disassembly and accelerates GBM growth, thus indicating that cAMP-induced DDX6 ubiquitylation represents a negative constraint on protein translation and cancer cell growth. This is consistent with the inhibitory role of GPCR-cAMP signaling on GBM cell growth (Safitri et al, 2020; Daniel et al, 2016; Warrington et al, 2010). The integration of cAMP signaling cascade and the ubiquitin system at P-bodies, thus, confers metabolic plasticity to cancer cells that, in response to metabolic needs, hormone stimulation or drug treatment, dynamically regulate protein translation and proliferative pathways. The apparent discrepancy between the inhibitory effects of praja2KO and those observed expressing the DDX6 ubiquitylation mutant on tumor growth might be explained by considering the more general role of praja2 in tumor cell biology (global translation, mitogenesis, metabolism, survival and growth control) that cannot be recapitulated by DDX6 allK/R mutant. In fact, expression of DDX6 mutant deregulates P-body formation and, probably, translation of selected RNA transcripts that may, thus, contribute to tumor cell growth.

In summary, we identified a cAMP-regulated ubiquitin network operating at the forefront of hormone signaling and P-bodies that dynamically regulates protein translation in GBM cells. This regulatory system may, thus, contribute to the adaptive metabolic rewiring underlying cancer growth, dissemination, and drug resistance.

## Methods

**Reagents and tools table**

| Reagent/resource | Reference or source | Identifier or catalog number |
|---|---|---|
| **Experimental models** | | |
| HEK293 | ATCC | |
| HeLa | ATCC | |
| U87MG | ATCC | |
| U87MG Praja2KO | This study | |
| U87MG Praja2KO +flag-praja2 | This study | |
| GBM primary cell line 3 | Pagano et al, 2021 | |
| GBM primary cell line 4 | Pagano et al, 2021 | |
| U87MG_luc2 | ATCC | |
| **Recombinant DNA** | | |
| DDX6-myc | Rouya et al, 2014 | Dr. Sonenberg |
| Gfp-DDX6 | Addgene | #25033 |

| Reagent/resource | Reference or source | Identifier or catalog number |
|---|---|---|
| DDX6 lysin mutant-myc | Genscript | N/A |
| flag-praja2 | Senatore et al, 2021 | N/A |
| flag-praja2rm | Senatore et al, 2021 | N/A |
| flag-praja2dm | Chiuso et al, 2023 | N/A |
| Flag-praja2D530-630 | Senatore et al, 2021 | N/A |
| HA-ubiquitin | Senatore et al, 2021 | N/A |
| **Antibodies** | | |
| Mouse anti-Gfp | Roche | #11814460001 |
| Mouse anti-flag | Merck | #F3165 |
| Mouse anti-GST | Santa Cruz Biotechnology | #sc-138 |
| Rabbit anti-praja2 | Bethyl | #A302-991A |
| Mouse anti-myc | Merck | #M4439 |
| Rabbit anti-myc | Abcam | #ab9106 |
| Rabbit anti-DDX6 | Novus biological | #NB200-192 |
| Mouse anti-DDX6 | Merck | #SAB4200837 |
| Mouse anti-HA.11 | Biolegend | #16B12 |
| Rabbit anti-K63 linkage specific polyubiquitin | Cell signaling | #5621 |
| Rabbit anti-K48 linkage specific polyubiquitin | Cell signaling | #4289 |
| Mouse anti-Hsp90 | Proteintech | #60318-1-Ig |
| Mouse anti-a-tubulin | Merck | #T5168-2 |
| Mouse anti-DCP1A | Merck | #WH0055802M6 |
| Rabbit anti-EDC3 | Proteintech | #16486 |
| Rabbit anti-TF | Proteintech | #17435-1-AP |
| Rabbit anti-CTNNA3 | Proteintech | #13974-1-AP |
| Rabbit anti-p21 | Proteintech | #10355-1-AP |
| Rabbit antic-myc agarose beads | Merck | #A7470 |
| IgG Rabbit | Invitrogen | #02-6102 |
| Goat anti-rabbit IgG-HRP | Biorad | #1706515 |
| Goat anti-mouse IgG-HRP | Biorad | #1706516 |
| **Oligonucleotides and other sequence-based reagents** | | |
| siRNA anti-praja2 | Dharmacon | #Q-006916-00-0020 |
| siRNA negative control | Ambion | # am4637 |
| Real-time primers for ADAMTS20 | IDT | Fw: 5'-AATCGTCCTGAGCCAAGAAAGG-3' Rev: 5'-GGAAGCCACCTCACATTAGAGG-3' |

| Reagent/resource | Reference or source | Identifier or catalog number |
|---|---|---|
| Real-time primers for CNKSR3 | IDT | Fw:5'-TCTCCTGCAGACAGATCTCAG-3' Rev: 5'-AGGGGAGCAGGAGTAAAGTTG-3' |
| Real-time primers for CLCA2 | IDT | Fw:5'-AGATGTGCAGCCTCAGAAGT-3' Rev: 5'-CTGCGGCTTGTTGTAGTTGA-3' |
| Real-time primers for ATR | IDT | Fw: 5'-CGGGAAATACTAGAACCTCA-3' Rev: 5'-CATAATGCTACAGCAGGTGA-3' |
| Real-time primers for USP9X | IDT | Fw: 5'-ACATGAGTCGCCTCCACCTG-3' Rev: 5'-GCCTGGGTGCACAGTCTTG-3' |
| Real-time primers for DHX9 | IDT | Fw: 5'-CGGTATGGAGATGGTCCACG-3' Rev: 5'-CGCTACCATAGCCTCCACTG-3' |
| Real-time primers for CTNNA2 | IDT | Fw: 5'-AATTAGCTCGTGCTGTGGCT-3' Rev: 5'-ACCGTGAGGACAACAGCATT-3' |
| Real-time primers for EMILIN1 | IDT | Fw: 5'-CAGCCTCTACACAGGTTCCAG-3' Rev: 5'-CACGTAGGCACACCAGTTCC-3' |
| Real-time primers for LUC | IDT | Fw: 5'-TACAACACCCCAACATCTTCGA-3' Rev: 5'-GGAAGTTCACCGGCGTCAT-3' |
| **Chemicals, enzymes, and other reagents** | | |
| forskolin | Merck | #F3917 |
| KT2059 | Merck | #K3761 |
| PGE2 | Merck | #P0409 |
| MG132 | Merck | #C2211 |
| TRI-Reagent | Merck | #T9424 |
| Dynabeads™ Protein A | Thermo Fisher Scientific | #10001D |
| TurboDNase | Thermo Fisher Scientific | #AM2239 |
| Proteinase K | Thermo Fisher Scientific | #AM2548 |
| DMEM | Gibco | #11965092 |
| L-Glutamine | Gibco | #25030-024 |

| Reagent/resource | Reference or source | Identifier or catalog number |
|---|---|---|
| FBS | Gibco | #A5256701 |
| Pen/Strep | Gibco | #15070-063 |
| DAPI | Thermo Fisher Scientific | #D1306 |
| Protein A/G plus Agarose | Santa Cruz | #sc-2003 |
| **Software** | | |
| Riborex with R studio version 4.1.0. | | |
| GraphPad | https://www.graphpad.com | |
| inBio Discover | | |
| STRING | Li et al, 2017 | |
| DESeq2 | Love et al, 2014 | |
| EnrichR | Kuleshov et al, 2016 | |
| IPA | www.qiagenbioinformatics.com/products/ingenuitypathway-analysis | |
| **Other** | | |

## Cell culture

Human embryonic kidney 293 (HEK293) cells, HeLa cells, human glioblastoma cells (U87MG), U87MG praja2KO, U87MG praja2KO+flag-praja2, and glioblastoma primary cell lines were cultured in Dulbecco's Modified Eagle Medium (DMEM) supplemented with 10% fetal bovine serum (FBS), 1% L-glutamine, 1% penicillin/streptomycin at 37 °C, 5% $CO_2$, and 95% humidity. For U87MG praja2KO+flag-praja2 cells, puromycin (1 μg/μl) was added. HEK293, U87MG, and HeLa cells were obtained from the American Type Culture Collection (ATCC). GBM3 and GBM4 primary glioblastoma cell lines were previously described (Pagano et al, 2021). U87MG praja2KO cells were produced at Biogem.

## Generation of U87MG praja2KO

CRISPR/Cas9 genome editing technology was performed by using the LenticrisprV2 plasmid to knockout praja2. Short guide RNA (sg RNA) against ATF6 (0.1 pmoL/μL sg RNA pairs 5′-AATGGACCAA-GAATCTGGTA-3′ and 5′-TACCAGATTCTTGGTCCATT-3′) was introduced to the Lenticrisprv2 plasmid through digestion/ligation cycles using BsmbI andT4 ligase (Neb). Genetic ablation of praja2 was confirmed by DNA sequencing.

Briefly, HEK293 cells were maintained in DMEM Complete medium (EuroClone, Pero (MI), Italy) supplemented with 10% FBS (Merck), 1% penicillin/streptomycin (Gibco), and 1% L-glutamine (Gibco), and used as packaging cells for lentiviral production. HEK293T cells were transfected by calcium-phosphate, using a DNA mix containing LenticrisprV2 (10 μg) harboring or not anti-praja2 human specific gRNA and psPAX2 (Addgene #12260; 18 μg) and pMD2.G (Addgene #12259; 13 μg) as packaging vectors. Forty-eight hours post-transfection, cell supernatants were collected, filtered through a 0.45-μM filter, and 20 μg/mL of polybrene (Merck) was added. U87 cell media was then replaced with infective

media, and cells were incubated for 6 h. Later, the infective cell medium was replaced with normal medium and, after additional 24 h, 1 μg/mL of puromycin was added to start selection. T7 assay was performed to confirm the efficiency of gRNA.

## Generation of U87MG praja2KO+flag-praja2

HEK293T cells were grown in a humidified incubator, at 37 °C, with 5% $CO_2$. The cells were cultured using DMEM (41966-029; Gibco) with 10% FBS (10270-106; Gibco), 2 mM L-glutamine (25030-024; Gibco), and 1% penicillin/streptomycin (P0781; Sigma-Aldrich). The cells were dissociated with trypsin-EDTA solution (T4049, Sigma-Aldrich) when the cultures reached ~90% confluency. Cell cultures were regularly tested by PCR to assure that cell lines were free from Mycoplasma contaminations. For transient transfection, HEK293T cells (at ~70% confluency) were transfected with 10 μg DNA plasmids using X-tremeGENE 9 DNA Transfection Reagent (06365779001; Sigma-Aldrich) diluted with serum-free DMEM (41966-029; Gibco), to a concentration transfection reagent to the diluted DNA (3:1 ratio of reagent to DNA). Transient transfections were performed according to the manufacturer's instructions. DNA complex was incubated for 15 min at room temperature and then the complex was added to the cells in a dropwise manner. To produce lentiviral supernatant, HEK293T cells were co-transfected with either the pLenti-CMV-Puro-flag-praja2 plasmid, pMD2.G (VSVG envelope expressing plasmid) and pMDLg/pRRE, pRSV/REV (third generation lentiviral packaging plasmids). The supernatants containing the lentiviral particles were collected at 72 h after transfection, centrifuged, and filtered, with Lenti-X concentrator added at 1:4 dilution (Clontech, #631231), and each titrated for U87MG cell line for a multiplicity of infection at 0.3 (25%). The polyclonal cells were created through selection with blasticidin (InvivoGen). The plasmids were a gift from D. Trono (Addgene Plasmid #12251, Plasmid #12253; and Plasmid #12259).

## Plasmid and transfection

DDX6-myc vector was gently provided by Dr. Sonenberg. Gfp-DDX6 was purchased by Addgene (#25033). DDX6 lysine mutant vectors were purchased by Genscript. Praja2 plasmids and HA-ubiquitin were previously described (Senatore et al, 2021; Chiuso et al, 2023). siRNAs targeting praja2 were purchased from Dharmacon. Praja2 siRNA sequences are: sequence 1: 5′-GAGAUGAGUUUGAAGAGUU-3′; sequence 2: 5′-GGGAGAAAUUCCUUGGGUUA-3′; sequence 3: 5′-UGACAAAGAUGAAGAUAGU-3′; sequence 4: 5′-UCAGAUGAC-CUCUUAAUAA-3′. Control siRNA was purchased from Ambion (#am4637).

## Immunoprecipitation, pull-down assay, and western blot

Cells were harvested and lysed in a specific buffer (1% Triton, 150 mM NaCl, 50 mM Tris HCl pH 7.5, 1 mM EDTA) supplemented with phenylmethylsulfonyl fluoride (PMSF) and a cocktail of protease inhibitors (Roche). Only for ubiquitylation assays, cells were lysed with triple detergent buffer (1% NP-40, 150 mM NaCl, 50 mM Tris HCl pH 8, 0.1% SDS, 0.5% NaDOC) supplemented with PMSF and protease inhibitors. Lysates were incubated overnight with the agarose beads conjugated with the primary

antibody (myc) or with the primary antibodies. In this case, A/G plus protein (Santa Cruz # Sc-2003) was added for 45 min. For pull-down assays, lysates were incubated with GST-fused proteins conjugated on glutathione beads. Beads were washed three times in lysis buffer. Precipitates and a quote of lysates were loaded on SDS polyacrylamide gels, blotted on a nitrocellulose membrane, blocked with 5% milk in TBS-Tween 0.1% and incubated with the primary and secondary antibodies. Proteins were detected with ECL (Euroclone).

## Immunofluorescence assay

Cells were plated on coverglass, fixed with 3% paraformaldehyde for 20 min, permeabilized with 0.3% Triton for 5 min, and aspecific antibodies binding sites were saturated by adding 3% BSA. Cells were incubated with the specific primary antibodies and then with the fluorescent-conjugated secondary antibodies (Invitrogen 1:200) for 30 min. Nuclei were stained with DAPI (Invitrogen 1:500) for 15 min. Staining was visualized using a Zeiss LSM700 confocal microscope. Statistical analysis of the immunofluorescence assay was performed using GraphPad.

## β-galactosidase staining

U87MG and U87MG praja2KO cells were plated in 6-well plates. After 24 h, the staining was performed by using the senescence β-galactosidase staining kit (Cell signaling; #9860) following the manufacturer's instructions.

## Electron microscopy

Cells were fixed with 2% formaldehyde and 0.2% glutaraldehyde in PHEM buffer (0.1 M), embedded in 12% gelatin, cooled on ice, and cut into 1-mm$^3$ cubes, at 4 °C. The cubes were kept in 2.3 M sucrose, at 4 °C, overnight, and then frozen in liquid nitrogen. Sixty-nanometer sections were cut with a diamond knife. The sections were picked up in a mix of 2% methyl cellulose and 2.3 M sucrose (1:1), and collected on grids covered with Formvar carbon supporting film (Electron Microscopy Sciences, PA, USA). The grids were incubated with the rabbit anti-praja2 antibody (#A302-991 A, Bethyl 1:15 dilution) and then mixed with different sizes of Protein A gold (10 nm) to reveal antigen staining. After labeling, the sections were treated with 1% glutaraldehyde and embedded in methylcellulose uranyl acetate for 10 min on ice. The excess of methyl cellulose uranyl acetate was removed, and the sections were dried at room temperature before their analysis at 120 kV in a Philips Tecnai 12 Biotwin electron microscope (FEI, Eindhoven, The Netherlands) using a VELETA digital camera.

## Protein–protein interaction network analysis

Immunoprecipitation and mass spectrometry analysis of praja2-interacting proteins was previously described (Delle Donne et al, 2022). praja2-interacting proteins identified in two different experimental conditions (see Dataset EV2) were analyzed using the inBio Discover and STRING Web tools (Li et al, 2017; Szklarczyk et al, 2023) inBio Map Data Version 2021_04_07) to build a protein–protein interaction (PPI) network based on highly trusted interactions derived from experimental evidence as well as

pathways and other curated resources. The following setting was applied for inBio Discover: Network Expansion disabled. The following settings were applied for STRING: "Network Type: Physical subnetwork" (the edges indicate that the proteins are part of a physical complex), "Active Interaction Sources: textmining, experiments and databases", "Minimum Required Interaction Score: 0.700". The praja2-PPI network was represented as an undirected graph (i.e., nodes and edges symbolize proteins and interactions, respectively). The network functional enrichment analysis was performed using two available methods: inBio Discover "Top 5 Pathways" (REACTOME, *Homo sapiens*) and STRING "Selected Pathways" (including REACTOME and Gene Ontology terms associated with RNA metabolism and processing).

## Gene-set enrichment analysis on U87MG

U87MG cells were transfected with siRNA targeting endogenous praja2 or with control siRNA for 48 h. Media was aspirated, plates were immediately placed on ice and washed with ice-cold phosphate-buffered saline (PBS). Total RNA extraction was performed using the TRI reagent (Merck #T9424) following the manufacturer's instruction. Total RNA concentrations were determined by using NanoDrop spectrophotometer (Thermo Fisher), and their quality was assessed with the TapeStation 4150 (Agilent Technologies). Briefly, 100 ng of RNA of each sample was subjected to mRNAs capture by using polyT-coated beads followed by the indexed library preparation with the SureSelect XT HS2 mRNA Library Preparation System (Agilent Technologies), according to the manufacturer's instructions. Finally, an equimolar amount of each DNA library was pooled together and subjected to cluster generation and sequencing into the Illumina NextSeq550 System (Illumina) with a 2 × 150 paired-end format. RNASeq FASTQ files were aligned to GRCh38 (GENCODE v36 p13) using the STAR 2-pass pipeline, following the Genomic Data Common procedure (Dobin et al, 2013). The STAR method was also considered for gene quantification. sipraja2 and siCNT U87MG cells gene expression were compared with DESeq2 (Love et al, 2014), resulting in the generation of the volcano plot in Fig. 1A. The recap of differential expression is the Wald's test statistics values (Dataset EV1a). Positive values were associated with the sipraja2 group; negative values were associated with the control group. The Heatmap in Fig. 1B shows the gene expression level (normalized as Transcript Per Million) of genes having the highest and lowest level in the differential gene profile. massiveGST (Frattini et al, 2018; Cerulo and Pagnotta, 2022) procedure provided the significant enrichment of the gene sets for the two groups from the MSigDB H, BP and MF collections (Liberzon et al, 2011). It was set up to consider significant enrichments only with an adjusted *q* value < 0.01 and corresponding to gene sets with a size of at least 50 genes (Dataset EV1d).

## Gene-set enrichment analysis on human tissue samples

Two scRNA-seq (GSE129671, (preprint: Ding et al, 2019; Data ref: Ding et al, 2019) 701 and 702) datasets, the GBM bulk RNAseq study from TCGA (Data ref: TCGA-GBM; phs000178), and the proteomic GBM profiles (Proteomic Data Common, ID: PDC000204 (Wang et al, 2021; Data ref: Wang et al, 2021) were analyzed. In addition, we used scRNA-seq data (GSE67835

(Darmanis et al, 2015; Data ref: Darmanis et al, 2015) from healthy adult brain donors. Given the different nature of the data, aiming to conduct consistent procedures for each dataset, it was considered a gene profile whose values were the Pearson linear correlation of praja2 with any other gene expression in the set (Dataset EV1a,b). These lists of genes fed the same procedure massiveGST to query the enrichment of the gene sets in the same collections considered in the gene-set enrichment analysis on U87MG. The significant results of each data set were intersected with gene sets enriched in sipraja2 and control cells to obtain the common macrocategories. To determine if the intersections were significant, we conducted a hypergeometric test following the methodology of the over-representation enrichment test (Draghici et al, 2003). Significant enrichments from the analysis of U87MG genomic profiles established two reference groups of gene sets: one associated with sipraja2 and the other with siCNT. Each in silico enrichment analysis from bulk RNAseq, scRNA-seq, proteomic and bulk RNAseq data of healthy brain tissue was compared with reference groups from U87MG, using the hypergeometric test to confirm the significance of the intersections. The $P$ values associated with the comparisons, ranging from $10^{-4}$ to $10^{-209}$, provided strong evidence for the significant intersection of the enrichment results. For example, gene sets showing a positive association with praja2 are over-represented with siCNT reference group with a $P$ value ~$10^{-112}$ (in the case of GSE129671_PW030_701). Comprehensive analyses are available in Appendix Table S1. The tabular results of enrichment analyses are in Dataset EV1d–i. The graphical rendering is in the Fig. 1 as network plots.

## Mass spectrometric analysis of DDX6 ubiquitylation and relative content

Strep-tactin purified protein Lysates from HEK293T cells transfected with DDX6-myc or empty vector ($n = 3$, biological replicates) were loaded on NuPAGE™ 4–12% Bis–tris acrylamide gels according to the manufacturer's instructions (Life Technologies). Running of protein was stopped as soon as proteins stacked in a single band. Protein-containing bands were stained with Imperial Blue (Pierce), cut from the gel, and digested with high sequencing grade trypsin (Promega) before mass spectrometry analysis. Briefly, gel pieces were washed and destained using few steps of 100 mM $NH_4HCO_3$. Destained gel pieces were shrunk with 100 mM ammonium bicarbonate in 50% acetonitrile and dried at RT. Protein spots were then rehydrated using 10 mM DTT in 25 mM ammonium bicarbonate pH 8.0 for 45 min at 56 °C. This solution was replaced by 55 mM iodoacetamide in 25 mM ammonium bicarbonate pH 8.0 and the gel pieces were incubated for 30 min at room temperature in the dark. They were then washed twice in 25 mM ammonium bicarbonate and finally shrunk by incubation for 5 min with 25 mM ammonium bicarbonate in 50% acetonitrile. The resulting alkylated gel pieces were dried at room temperature. The dried gel pieces were reswollen by incubation in 25 mM ammonium bicarbonate pH 8.0 supplemented with 12.5 ng/µl trypsin (Promega) for 1 h at 4 °C and then incubated overnight at 37 °C. Peptides were harvested by collecting the initial digestion solution and carrying out two extractions; first in 5% formic acid and then in 5% formic acid in 60% acetonitrile. Pooled extracts were dried down in a centrifugal vacuum system. Samples were reconstituted in 0.1% TFA 4% acetonitrile before mass spectrometry using an Orbitrap Fusion Lumos Tribrid Mass Spectrometer (ThermoFisher Scientific, San Jose, CA) online with an Ultimate

3000RSLCnano chromatography system (ThermoFisher Scientific, Sunnyvale, CA). Peptides were separated at 40 °C using a two steps linear gradient (4–20% acetonitrile/H2O; 0.1% formic acid for 110 min and 20–32% acetonitrile/H2O; 0.1% formic acid for 10 min). An EASY-Spray nanosource was used for peptide ionization (2200 V, 275 °C). MS was conducted using a data-independent acquisition mode (DIA). Full MS scans were acquired in the range of $m/z$ 375–1500 at a resolution of 120,000 at $m/z$ 200, and the automatic gain control (AGC) was set at $4.0 \times 10E5$ with a 50 ms maximum injection time. MS2 spectra were acquired in the Orbitrap with a resolution of 30,000, in the mass range of 200–1800 $m/z$ after isolation of parent ion in the quadrupole and fragmentation in the HCD cell under collision Energy of 30%. DIA parent ion range was from 400 to 1000 $m/z$ divided into 40 windows 16 Da wide and from 1000 to 1500 $m/z$ divided into 10 windows 50 Da wide.

For protein identification and quantification, relative intensity-based label-free quantification (LFQ) was processed using the DIA-NN 1.8 algorithm (Demichev et al, 2020). Raw files were searched against the human database from UniProt (20 404 entries, 2023/01/10) implemented with a contaminant database (Frankenfield et al, 2022). The following parameters were used for searches: (i) trypsin allowing cleavage before proline; (ii) one missed cleavage was allowed; (iii) cysteine carbamidomethylation ($+ 57.02146$) as a fixed modification and methionine oxidation ($+ 15.99491$) and N-terminal acetylation ($+ 42.0106$) as variable modifications; (iv) a maximum of 1 variable modification per peptide allowed; and (v) minimum peptide length was 7 amino acids and a maximum of 30 amino acids. The match between runs option was enabled. The precursor false discovery was set to 1%. DIA-NN parameters were set on double-pass mode for Neural Network classifier, Robust LC High precision for quantification strategy and RT-dependent mode for Cross-run normalization. Library was generated using Smart profiling set up. MS1 and MS2 mass accuracy was automatically calculated for precursor charge fixed between 2 and 4. The mass spectrometry proteomics data have been deposited to the ProteomeXchange Consortium via the PRIDE partner repository with the data set identifier PXD054750. Main output file from DIA-NN was further filtered at 1% FDR and LFQ intensity was calculated using our DIAgui package at 1% $q$ value (https://github.com/marseille-proteomique/DIAgui (Gerault et al, 2024)). The statistical analysis was done with Perseus software (version 1.6.15.0), where proteins were pre-filtered to remove contaminants, and those with less than 3 valid quantitative values across biological replicates. Missing values were then imputed using Perseus's normal distribution imputation method (Width: 0.3, Down shift: 1.8). Differential proteins were assessed using a two-sample $t$ test with a permutation-based false discovery rate of 1%, employing 250 iterations and the parameter So set to 0.1 (Tyanova and Cox, 2018).

## K-GG peptides identification

Raw files generated for affinity purification mass spectrometry were re-processed to identified and quantified K-GG peptides. The search was done similarly as above with the following modifications. A parent ion tolerance of 5 ppm and a fragment ion tolerance of 10 ppm was set. Dynamic K-GG modifications was added using two additional options (--var-mod UniMod:121, 114.042927,K and --monitor-mod UniMod:121). The mass spectrometry proteomics data have been deposited to the ProteomeXchange Consortium via the PRIDE partner repository with the data set identifier

PXD054750. Main output file from DIA-NN was filtered as previously and K-GG peptides identification and quantification extracted using our DIAgui package at 1% *q* value (https://github.com/marseille-proteomique/DIAgui (Gerault et al, 2024)). The same dataset as above was processed to identify and quantify K-GG peptides using the same parameters as above, with the exception of additional options to allow the identification of these modified peptides. The mass spectrometry proteomics data have been deposited to the ProteomeXchange Consortium via the PRIDE partner repository with the data set identifier PXD054750. Volcano plots have been generated by Prism 10 (GraphPad) by plotting for each protein the log2 fold change (*X* axis) and the −log10 of the *P* value (*Y* axis).

## Polysome profiling sequencing (PP-seq)

Media was aspirated from U87MG cells (two 150 mm plates per condition). Plates were immediately placed on ice and washed with ice-cold 1× PBS containing cycloheximide (100 µg/mL, Sigma-Aldrich). Cells were lysed in 800 µL of polysome extraction buffer (10 mM Tris HCl pH 7.4, 100 mM KCl, 10 mM MgCl2, 1% Triton X-100, 1 mM DTT, 10 U/mL RNaseOUT [Invitrogen], 100 µg/mL cycloheximide) and incubated on ice for 10 min. Lysates were cleared by centrifugation at 14,000 rpm for 10 min, at 4 °C. The supernatant was collected, and the corresponding absorbance was measured at 260 nm using a NanoDrop. Between eight and ten optical density units were loaded onto a 10–60% sucrose gradient formed by layering 6 mL of 10% sucrose over 6 mL of 60% sucrose prepared in polysome extraction buffer without Triton X-100 and containing 0.5 mM DTT, in a 12-mL tube (Polyallomer; Beckman Coulter). Gradients were prepared using a gradient maker (Gradient Master; Biocomp). Polysomes were separated by centrifugation at 37,000 rpm for 2 h using a Beckman SW41 rotor. Twelve fractions of 920 µL were collected, and polysomes were monitored by measuring absorbance at 254 nm. RNA extraction was performed by adding TRI reagent (Merck #T9424) at a 1:1 v/v ratio to the fractions, according to the manufacturer's protocol. Total RNA concentration values were determined by using a NanoDrop spectrophotometer (Thermo Fisher) and their quality were assessed with the TapeStation 4150 (Agilent Technologies). Briefly, 150 ng of RNA of each sample were subjected to mRNAs capture by using polyT-coated beads followed by the indexed library preparation with the SureSelect XT HS2 mRNA Library Preparation System (Agilent Technologies), according to the manufacturer's instructions. Finally, an equimolar amount of each DNA library was pooled together and subjected to cluster generation and sequencing into the Illumina NextSeq550 System (Illumina) with a 2 × 75 paired-end format. Differential expression analysis to identify differentially translated genes was conducted using Riborex with R studio version 4.1.0.

## RNA-immunoprecipitation sequencing (RIP-seq)

Media was aspirated from U87MG cells (two 150 mm plates per condition). Plates were immediately placed on ice and washed with ice-cold 1X PBS. Cells were lysed in 1 mL of RNA-immunoprecipitation (RIP) buffer (20 mM Tris-Cl pH 7.4, 150 mM NaCl, 1 mM EDTA, 0.5% NP-40, 1 mM DTT) containing 1:1000 protease inhibitor (Sigma) and 100 U of RNase inhibitor (20

U/µL, Thermo Fisher Scientific) on ice, for 30 min. Cell lysate was cleared by centrifugation at 15,000 × g for 10 min, at 4 °C. Subsequently, the supernatant was collected, and protein concentration was measured. Three mg of extract were used for each IP in a final volume of 1 mL. Five µg of DDX6 antibody and 5 µg of anti-IgG antibody (as a control) were added to the supernatant and incubated for 2 h, at 4 °C, under gentle rotation. Fifty µL of Dynabeads™ Protein A (10001D, Thermo Fisher Scientific) were washed twice with 0.5 mL of RIP buffer, added to each sample, and then incubated for 2 h at 4 °C. The beads were collected and washed three times with 0.5 mL of RIP buffer; the last wash was performed with 0.5 mL of 1× PBS. Next, beads were resuspended in 0.1 mL of 1× PBS, followed by DNA digestion with 1 µL of TurboDNase (AM2239, Thermo Fisher Scientific) for 15 min, at 37 °C, at 1100 rpm. Then, 0.1% SDS (Sigma) and 50 µg proteinase K (20 mg/mL, AM2548, Thermo Fisher Scientific) were added to each sample and incubated for 30 min, at 50 °C, with 1100 rpm. DNA- and protein-free RNAs were collected from the beads using a magnetic stand. RNAs were further purified through RNA Clean and Concentrator-5 (R1013, Zymo Research) and eluted in an 8 µL-volume. Purified RNAs were used to make the cDNA libraries. Briefly, 150 ng of RNA of each input sample were used for library preparation as previously described, while an equal volume (5 µl) of each IP-purified RNAs were processed using the SureSelect XT HS2 mRNA Library Preparation System kit (Agilent Technologies), according to the manufacturer's instructions. The resulting cDNA libraries were quantified by using the Qubit dsDNA BR Assay (Thermo Fisher) and analyzed by using the TapeStation chip (Agilent Technologies). Finally, in the case of input libraries, an equimolar amount of each library was pooled together, while in the case of the libraries prepared from the IP-samples a 4 nM pool were done loading an equal volume of each DNA library. In both cases, RNA sequencing was finally carried out on an Illumina NextSeq550 instrument with a 2 × 75 paired-end format. For RIP analysis, we considered enriched all mRNAs identified in IP-samples that presented a fold enrichment > 2 compared to input samples with an adjusted *P* value < 0.05. Furthermore, we excluded all enriched transcripts that displayed enrichment in control samples (NO antibody) compared to the input with cutoffs mentioned above (Tarallo et al, 2017). For DDX6 target validation, RIP was carried out as above. Before TurboDNase treatment, 5 µL of in vitro transcribed luciferase mRNA (5 pg/µL) were added to DDX6 and IgG eluates. RNAs were further purified through RNA Clean and Concentrator-5 (R1013, Zymo Research) and eluted in a 10 µL-volume. Five µL of RNA were reverse transcribed using Super-ScriptIII (18080093, Thermo Fisher Scientific) following the manufacturer's instructions. The resulting cDNA was used to amplify selected mRNA targets. For all gene ontology analysis, the enrichR webtools was used (Kuleshov et al, 2016). For the analysis of the input, data were analyzed through DESeq2 and IPA (QIAGEN Inc., https://www.qiagenbioinformatics.com/products/ingenuitypathway-analysis).

## Xenograft models of glioblastoma

For in vivo experiments with U87MG praja2KO cells, we used male CD1 nude mice (20–22 g body weight) (Charles River, Calco, CO, Italy) with a 20–22 g body weight, developed from the transfer of the nude gene from Crl:NU-Foxn1nu to a CD-1® mouse through a

series of crosses and backcrosses beginning in 1979 at Charles River Laboratories, Wilmington, MA. The animals lack thymus and are unable to produce T-cells, and is therefore immunodeficient. Male mice were anesthetized with ketamine (100 mg/kg, intraperitoneal) and xylazine (10 mg/kg, intraperitoneal) and injected with U87MG cells and U87MG praja2KO subcutaneously implanted on the left side of their flank (n° 711/2023-PR, prot. 30234.90). Implanting cells at a rate of 1 μl/min of infusion resulted in $5 \times 10^6$ cells/200 μl.of DMEM with matrigel (Corning). The volume of subcutaneous tumors was assessed every week with a caliper. Forty-two days after injection, mice were sacrificed, and the isolated tumor masses underwent to a cycle of paraffin embedding. Serial sections with a thickness of 5 μm were cut from the beginning of each tumor. The sections were then submitted to immunohistochemistry after being stained with Mayer's hematoxylin and eosin (both from Diapath, Bergamo, Italy).

For in vivo experiments with DDX6 infected cells, the study was performed in the Test Facility of BIOGEM, authorized to carry out this experimentation by means of the Ministerial Authorization n° 257/2023-PR and authorized for the use of animal for scientific purpose and regulatory research according to the Italian Decree N° 26/2014: authorization N° 08/2023-UT del 23/03/2023. A total of 18 mice nude female CD1 (24–25 g body weight; Charles River, Calco, CO, Italy) were housed in controlled environments with a 12-h light/dark cycle, 22 °C temperature, and 55% humidity. Food and water were provided freely. Mice were given water on a free-choice basis along with normal rodent pelleted chow 4RF21 (Mucedola, Settimo Milanese, Italy) (5 weeks-old, $N = 6$/group) were injected subcutaneously in the right flank with 2.5×106/200 μl U87MG_luc2 cells (from ATCC) infected with empty virus or the virus carrying DDX6-myc or DDX6 allK/R-myc cells. The injected cells were resuspended in Matrigel (Geltrex™ LDEV-Free Reduced Growth Factor Basement Membrane Matrix). The welfare of the animals was assessed daily and any clinical signs of disease or other clinical or behavioral abnormalities were reported. Subcutaneous tumor growth was assessed twice weekly by caliper measurement and weight was recorded once weekly. Approximately 46 days after the injection, animals were sacrificed, and tumor masses were collected and stored in 10% buffered formalin for analysis.

## Immunohistochemistry

The immunohistochemical analysis for praja2, Ki67, p21, and p53 was carried out on sliced from subcutaneously tumor from control group (mice injected with U87MG) and praja2KO (mice injected with U87MG praja2KO). Slides were fixed with 70% ethanol and immunostained using a Benchmark Ultra XT (Roche) for the proliferation marker ki67 (prediluted from Roche), praja2 (1:200, Novus Biological), p21 (1:100, Proteintech), and p53 (1:100, Proteintech). The ultraview DAB detection kit (Roche) was used to detect antibodies. Only secondary antibodies were used to incubate the negative control samples. Each tumor's representative slide was stained also with hematoxylin and eosin.

## Ethical approval

Animal experiments were performed under the following approval protocol: n° 711/2023-PR, prot. 30234.90 and Italian Decree N° 26/

2014: authorization N° 08/2023-UT of 23/03/2023 (Ministerial Authorization n° 257/2023-PR).

## Data availability

RNAseq raw data are publicly available in ArrayExpress under accession numbers: E-MTAB-14326 (https://www.ebi.ac.uk/biostudies/arrayexpress/studies/E-MTAB-14326?key=17472c0e-3d95-4684-ae06-9ed4d7a1b39e) for RNAseq, E-MTAB-14328 (https://www.ebi.ac.uk/biostudies/arrayexpress/studies/E-MTAB-14328?key=9d730938-69c5-4ff5-b3b0-39a34497b2f8) for PP-seq, E-MTAB-14331 (https://www.ebi.ac.uk/biostudies/arrayexpress/studies/E-MTAB-14331?key=853298a9-5e4f-4079-b137-d93fc35efcfd) for RIP-seq. The mass spectrometry-based proteomic data have been deposited to the ProteomeXchange Consortium via the PRIDE partner repository with the dataset identifier PXD034966 for praja2 MS and PDX054750 for DDX6 ubiquitylation MS. Source data are available.

The source data of this paper are collected in the following database record: biostudies:S-SCDT-10_1038-S44319-025-00425-5.

## Peer review information

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

## Acknowledgements

The authors thank Nicola Russo, Luca Roberto, and Filomena Russo for the generation of U87MG praja2KO cells and infections of GBM cells with lentiviral particles encoding for DDX6 variants at the Gene Service Facility of Biogem, Biology and Molecular Genetics Institute, Ariano Irpino, Pasqualino De Antonellis for infections of U87MG praja2KO cells with lentiviral particles encoding for praja2, Prof. Giuseppe Merla and Paolo Salerno for their support at RNAseq Facility at DMMBM, Dr Seetharaman Parashuraman for support in Electron Microscopy analysis. Proteomics analyses were done using the mass spectrometry facility of Marseille Proteomics (marseille-proteomique.univ-amu.fr). Intomics A/S has not endorsed our research results or our published article. This paper is dedicated to the memory of Giorgio Giurato, a great friend and remarkable colleague. This work was supported by Fondazione AIRC per la Ricerca sul Cancro (IG2023-29124), the Italian Ministry of University and Research (National Center for Gene Therapy and Drugs based on RNA Technology, PNRR-CN3: E63C22000940007; PRIN2022: E53D23009690006 and E53D23021760001, MNESYS-A multiscale integrated approach to the study of the nervous system in health and disease' PE0000006, DN.1553), European Regional Development Fund (POR Campania FESR 2021-2027), grant "RARE.Glials" to AF. Proteomics analyses were supported by IBISA, the Cancéropôle PACA, the Provence-Alpes-Côte d'Azur Region, the Institut Paoli-Calmettes, and Fonds Européen de Développement Regional (FEDER).

## Author contributions

**Emanuela Senatore**: Conceptualization; Data curation; Investigation; Methodology; Writing—original draft; Writing—review and editing. **Rosario Avolio**: Data curation; Formal analysis; Investigation; Writing—original draft. **Laura Rinaldi**: Investigation. **Francesco Chiuso**: Investigation. **Maria A Oliva**: Investigation. **Chiara D'Ambrosio**: Data curation; Formal analysis. **Antonio Giuseppe Bianco**: Investigation. **Emiliano Dalla**: Data curation; Formal analysis.

**Stefano Maria Pagnotta**: Data curation; Formal analysis. **Raffaella Flammia**: Data curation; Formal analysis. **Concetta Ambrosino**: Investigation. **Domenico Memoli**: Data curation; Formal analysis. **Gabriele Turacchio**: Investigation. **Sonia Ines Mimoune**: Data curation; Formal analysis. **Yves Toiron**: Data curation; Formal analysis. **Stephane Audebert**: Data curation; Formal analysis. **Luc Camoin**: Data curation; Formal analysis. **Luca Lignitto**: Data curation; Formal analysis. **Andrea Scaloni**: Data curation; Formal analysis; Writing—review and editing. **Antonietta Arcella**: Data curation; Investigation. **Antonio Feliciello**: Conceptualization; Resources; Supervision; Funding acquisition; Writing—original draft; Project administration; Writing—review and editing.

Source data underlying figure panels in this paper may have individual authorship assigned. Where available, figure panel/source data authorship is listed in the following database record: biostudies:S-SCDT-10_1038-S44319-025-00425-5.

## Disclosure and competing interests statement

The authors declare no competing interests.

# Expanded View Figures

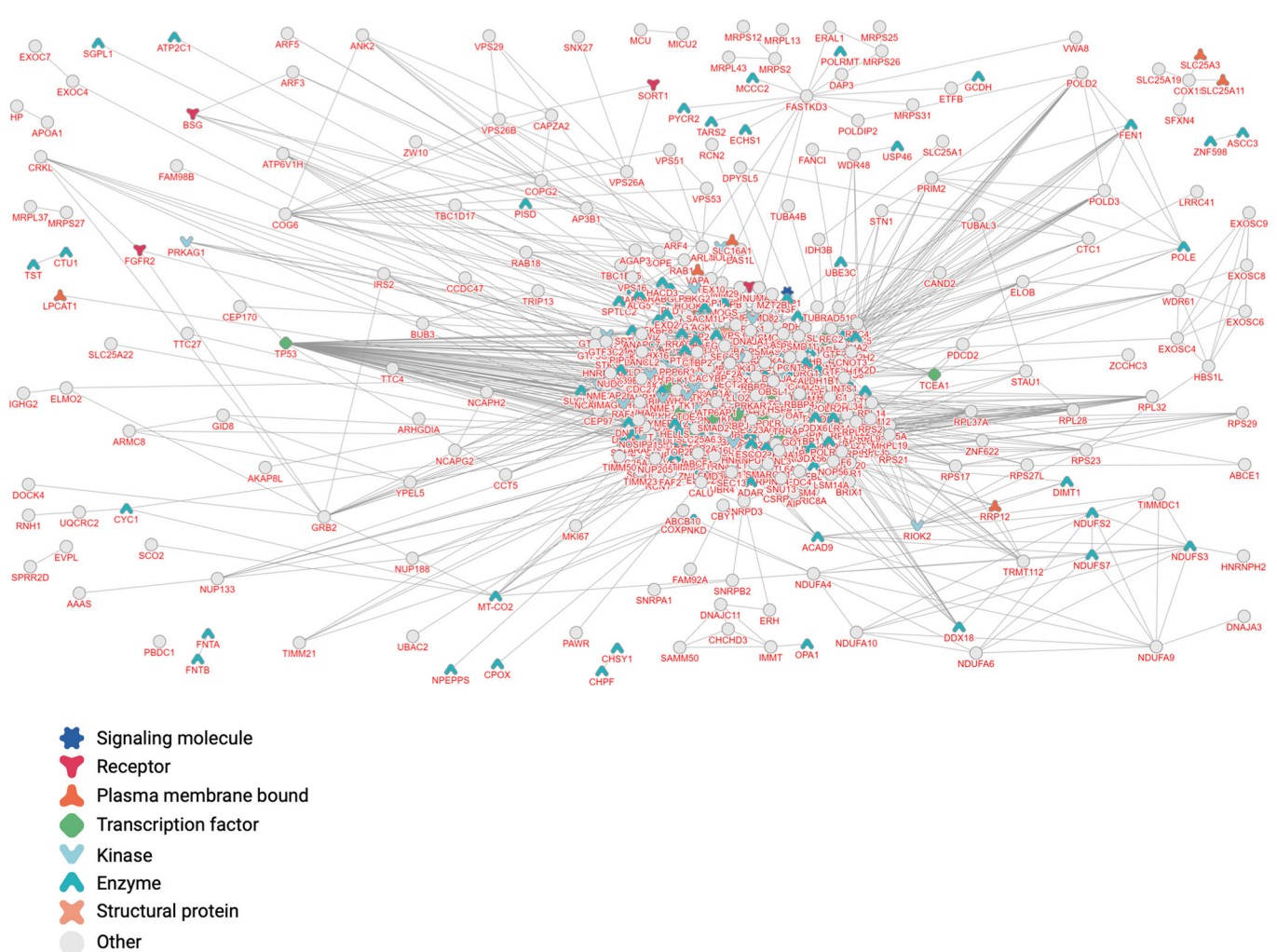

**Figure EV1. Praja2 protein–protein interaction network.**

Protein–protein interaction network of all the directly-interacting components belonging to the praja2 interactome, which was generated using the inBio Discover web tool (no network expansion) based on highly trusted interactions ($n = 506$ proteins, $n = 1563$ interactions). The figure legend indicates the functional roles of the displayed interactors.

**A**

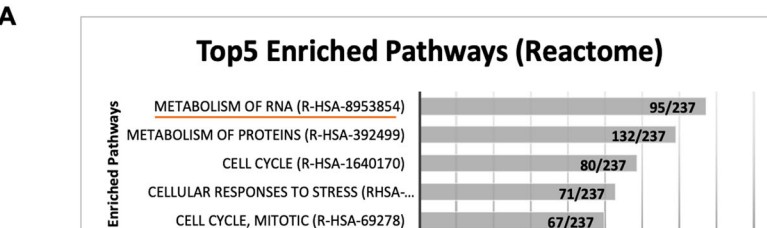

**B**

Figure EV2.    Praja2 interacts with components of metabolism of RNA pathway.

(A) Barplot showing the Top5 most enriched REACTOME terms derived from the analysis of praja2-PPI network. (B) Protein–protein interaction network of components involved in cell cycle, response to stress and metabolism, which was generated using the inBio Discover web tool (no network expansion) querying the Reactome database for functional enrichment. The interactions existing between proteins involved in cell cycle (blue nodes), cell cycle, mitotic (cyan nodes), cellular responses to stress (green nodes), metabolism of proteins (purple nodes) and metabolism of RNA (magenta nodes) are shown ($n = 237$ proteins, $n = 762$ interactions).

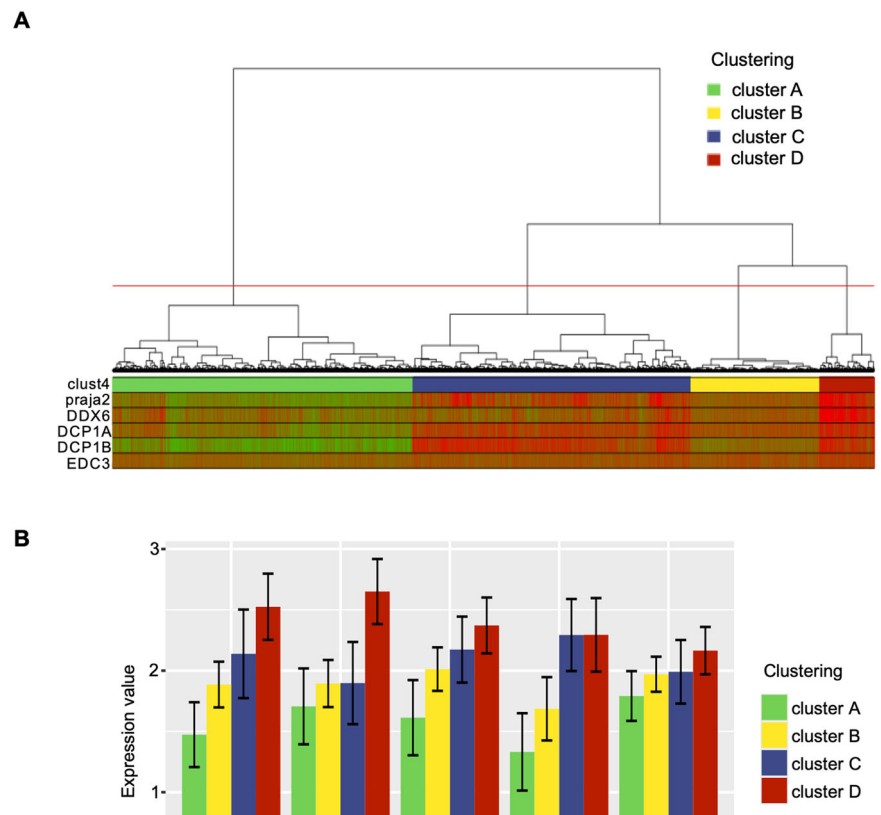

**Figure EV3. Praja2 is a potential regulator of mRNA translation.**

(A) Unsupervised hierarchical clustering of 20.000 cells from IDH wild-type GBM donor 701 (GSM3719277). The first annotation bar is the primary statistical clustering, while other bars show expression levels of the indicated genes (green= low expression, red= high expression). (B) Gene expression level across clusters. The bar graphs show each cluster's mean of gene expression levels of praja2, DDX6, DCP1A, DCP1B, and EDC3. Data from 20000 cells were used. Values represent the mean ± SD.

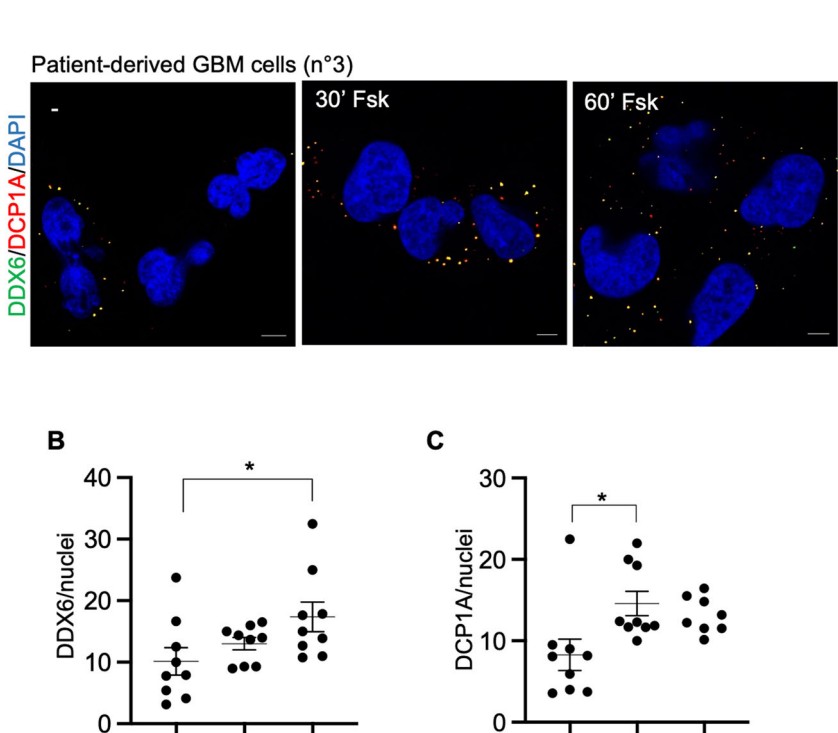

**Figure EV4. cAMP induces P-body formation.**

(**A**) GBM cells derived from patient 3 were treated 30 and 60 min with Forskolin (10 µM). Cells were fixed and immunostained with anti-DDX6, anti-DCP1A and DAPI. Scale bar: 5 µm. (**B, C**) Quantitative analysis of three biological independent experiments, mean ± SEM is indicated. *t* test *$P < 0,05$ (**B**, $P = 0.0413$; **C**, $P = 0.0213$). Source data are available online for this figure.

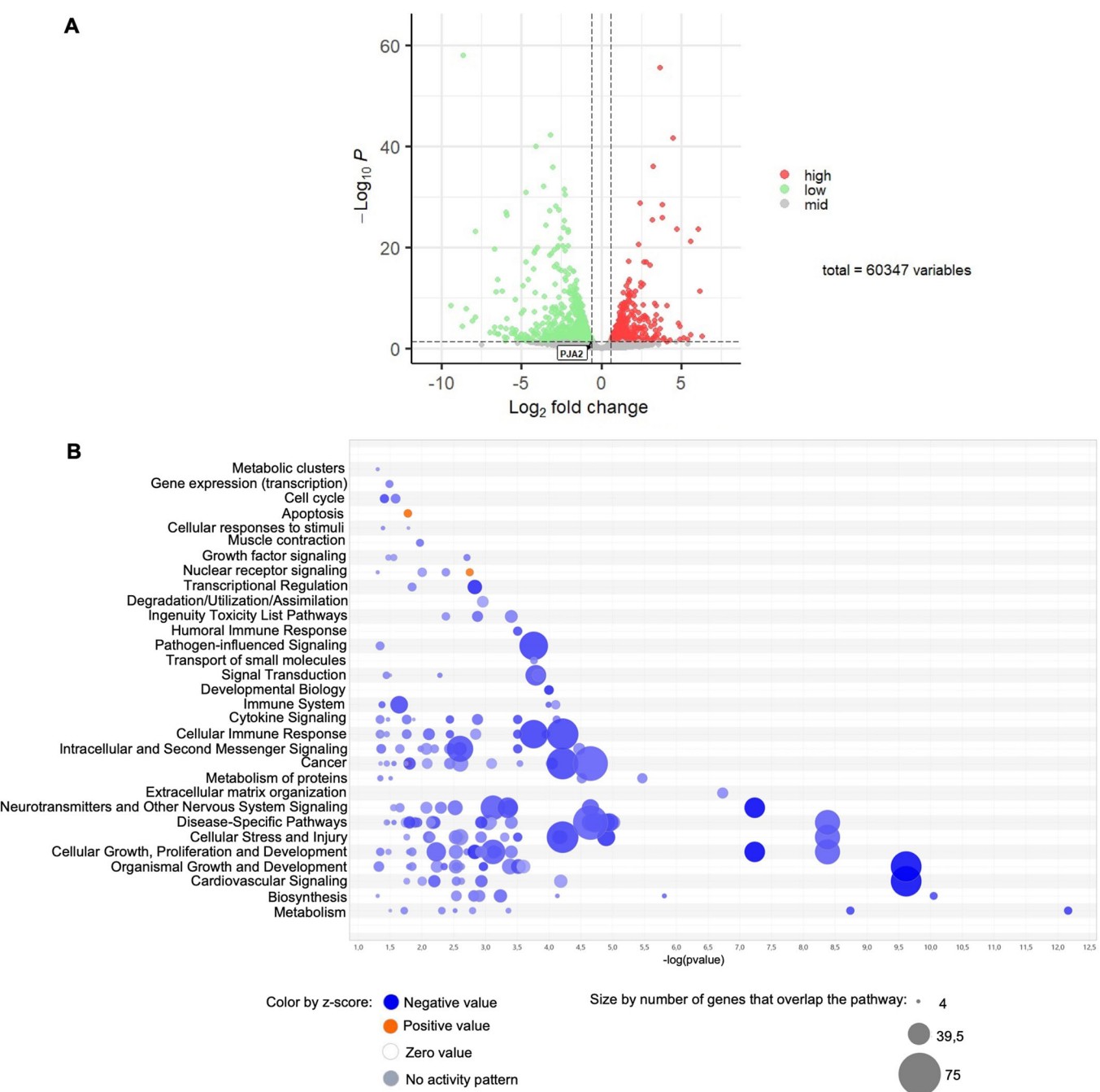

**Figure EV5. Praja2 regulates transcriptome of GBM cells.**

(A) Volcano plot showing log2FC and $P$ value adjusted distribution of all genes, in particular down and upregulated differentially expressed (DE) genes comparing praja2KO versus WT cells are presented in green and red, respectively ($n = 3$ independent biological replicates). The statistical analysis was performed with Wald test of Deseq2. (B) Enrichment results of Ingenuity Pathways Analysis on DE genes, the size of the bubble indicates the number of enriched genes for each term. Statistical analysis performed with Fisher's Exact Test.

