## [Peer Review File · EMBO Reports]

Praja2 controls P-body assembly and translation in glioblastoma by non-proteolytic ubiquitination of DDX6

Emanuela Senatore, Rosario Avolio, Laura Rinaldi, Francesco Chiuso, Maria A. Oliva, Chiara D'Ambrosio, Antonio Giuseppe Bianco, Emiliano Dalla, Stefano Maria Pagnotta, Raffaella Flammia, Concetta Ambrosino, Domenico Memoli, Gabriele Turacchio, Sonia Ines Mimoune, Yves Toiron, Stephane Audebert, Luc Camoin, Luca Lignitto, Andrea Scaloni, Antonietta Arcella and Antonio Feliciello

Corresponding author(s): Antonio Feliciello (feliciel@unina.it)

Review Timeline:

Submission Date:	8th Oct 24
Editorial Decision:	21st Nov 24
Revision Received:	20th Dec 24
Editorial Decision:	11th Feb 25
Revision Received:	17th Feb 25
Accepted:	27th Feb 25

Transaction Report:

Dear Prof. Feliciello

Thank you for the submission of your research manuscript to our journal. We have now received the two enclosed reports on it.

As you will see, both referees acknowledge that the findings are potentially interesting but they also raise a number of concerns and have suggestions how to further strengthen the data that should be addressed.

Given these constructive comments, we would like to invite you to revise your manuscript with the understanding that the referee concerns (as detailed above and in their reports) must be fully addressed and their suggestions taken on board. Please address all referee concerns in a complete point-by-point response. Acceptance of the manuscript will depend on a positive outcome of a second round of review. It is EMBO Reports policy to allow a single round of revision only and acceptance or rejection of the manuscript will therefore depend on the completeness of your responses included in the next, final version of the manuscript.

We realize that it is difficult to revise to a specific deadline. In the interest of protecting the conceptual advance provided by the work, we recommend a revision within 3 months (February 21st). Please discuss the revision progress ahead of this time with the editor if you require more time to complete the revisions.

I am also happy to discuss the revision further via e-mail or a video call, if you wish.

*****IMPORTANT NOTE:

We perform an initial quality control of all revised manuscripts before re-review. Your manuscript will FAIL this control and the handling will be delayed IN CASE the following APPLIES:

- 1) A data availability section providing access to data deposited in public databases is missing. If you have not deposited any data, please add a sentence to the data availability section that explains that.
- 2) Your manuscript contains statistics and error bars based on $n=2$. Please use scatter blots in these cases. No statistics should be calculated if $n=2$.

When submitting your revised manuscript, please carefully review the instructions that follow below. Failure to include requested items will delay the evaluation of your revision.*****

2) individual production quality figure files as .eps, .tif, .jpg (one file per figure).

Please download our Figure Preparation Guidelines (figure preparation pdf) from our Author Guidelines pages <https://www.embopress.org/page/journal/14693178/authorguide> for more info on how to prepare your figures.

4) a complete author checklist, which you can download from our author guidelines

(<<https://www.embopress.org/page/journal/14693178/authorguide>>). Please insert information in the checklist that is also reflected in the manuscript. The completed author checklist will also be part of the RPF.

5) Please note that all corresponding authors are required to supply an ORCID ID for their name upon submission of a revised manuscript (<<https://orcid.org/>>). Please find instructions on how to link your ORCID ID to your account in our manuscript tracking system in our Author guidelines

(<<https://www.embopress.org/page/journal/14693178/authorguide#authorshipguidelines>>)

6) We replaced Supplementary Information with Expanded View (EV) Figures and Tables that are collapsible/expandable online. A maximum of 5 EV Figures can be typeset. EV Figures should be cited as 'Figure EV1, Figure EV2' etc... in the text and their respective legends should be included in the main text after the legends of regular figures.

7) Before submitting your revision, primary datasets (and computer code, where appropriate) produced in this study need to be deposited in an appropriate public database (see <<https://www.embopress.org/page/journal/14693178/authorguide#dataavailability>>). <optional: Specifically, we would kindly ask you to provide public access to the the

The accession numbers and database should be listed in a formal "Data Availability " section (placed after Materials & Method) that follows the model below (see also <<https://www.embopress.org/page/journal/14693178/authorguide#dataavailability>>). Please note that the Data Availability Section is restricted to new primary data that are part of this study.

Data availability

Additional information on source data and instruction on how to label the files are available <<https://www.embopress.org/page/journal/14693178/authorguide#sourcedata>>.

10) Figure legends and data quantification:

- the name of the statistical test used to generate error bars and P values,
 - the number (n) of independent experiments (please specify technical or biological replicates) underlying each data point,
 - the nature of the bars and error bars (s.d., s.e.m.)
- If the data are obtained from n {less than or equal to} 5, show the individual data points in addition to the SD or SEM.
- If the data are obtained from n {less than or equal to} 2, use scatter blots showing the individual data points.

11) Our journal encourages inclusion of *data citations in the reference list* to directly cite datasets that were re-used and obtained from public databases. Data citations in the article text are distinct from normal bibliographical citations and should directly link to the database records from which the data can be accessed. In the main text, data citations are formatted as follows: "Data ref: Smith et al, 2001" or "Data ref: NCBI Sequence Read Archive PRJNA342805, 2017". In the Reference list, data citations must be labeled with "[DATASET]". A data reference must provide the database name, accession number/identifiers and a resolvable link to the landing page from which the data can be accessed at the end of the reference. Further instructions are available at <<https://www.embopress.org/page/journal/14693178/authorguide#referencesformat>>.

12) All Materials and Methods need to be described in the main text using our 'Structured Methods' format. According to this format, the Methods section includes a Reagents and Tools Table (listing key reagents, experimental models, software and relevant equipment and including their sources and relevant identifiers) followed by a Methods and Protocols section describing the methods, ideally using a step-by-step protocol format. The aim is to facilitate adoption of the methodologies across labs. Please download and fill our Reagents and Tools Table template (.docx), which you can find in our author guidelines: <https://www.embopress.org/page/journal/14693178/authorguide#structuredmethods>. When submitting your revised manuscript, please do not include the Reagents and Tools Table in the Methods section of the manuscript but upload it as a separate file choosing the file type "Reagent Table". An example of a Method paper with Structured Methods can be found here: <https://www.embopress.org/doi/10.15252/msb.20178071>.

13) As part of the EMBO publication's Transparent Editorial Process, EMBO Reports publishes online a Review Process File to accompany accepted manuscripts. This File will be published in conjunction with your paper and will include the referee reports, your point-by-point response and all pertinent correspondence relating to the manuscript.

Kind regards,

=====

Referee #1:

The authors convincingly show that DDX6 binds to and is ubiquitinated by the Praja2 protein, and this ubiquitination is cAMP dependent. The authors further show that praja2 is necessary for FSK-induced P-body formation and suggest an axis of cAMP-PKA-Praja2-DDX6 in the formation of P-bodies. Interestingly, the authors also show that Praja2 ko can induce premature cellular senescence in glioblastoma cells with in vitro and in vivo evidence. However, the authors describe other findings that disagree with prior research without sufficient comment (DDX6 increasing translation, reduced P-bodies accelerating tumorigenesis). These findings may be highly valuable to the field but would require special attention beyond what is present as they are counterintuitive to past research. An additional serious concern is that there are frequent claims in the initial sections that are made without supporting data or attribution to previous results. I have listed my specific comments below.

Minor Comments

1. P-bodies are not canonically considered organelles (line 80) but may be referred to as membraneless organelles
2. There are grammatical issues throughout the manuscript, most notably for plurals as in the case of P-bodies, (i.e. line 80) "P-bodies dynamics" should be "the dynamics of P-bodies" or "P-body dynamics"
3. Fig. 2C is a confusing presentation of this data as it applies a linear relationship between cluster A-D. A bar graph would be more suitable.

Major Comments

1. The data present in Fig 1 is insufficient for an entire figure as it is only gene ontology analysis. Since it is derived from numerous datasets, efforts should be made to include additional data from the RNA-seq analysis and patient data, not limited to the identity of genes that are regulated beyond just their ontological classification. The presentation of this analysis also should include false discovery rates and enrichment magnitudes instead of just a binary classification as it currently appears to represent. It is also unclear whether the patient data was performed by the authors or is obtained from publicly available sources; the latter case demands citations alongside those figure notations or in the legend. Finally, the conclusion of this section is specious as only a visually suggested similarity between ontological enrichment between patient samples and cell culture samples with siRNA against PRAJA2 is provided.
2. The data used to generate Fig S1A is not cited or discussed. If the authors performed an affinity purification of Praja2, they should show western blot validation. If it is the data of other studies, it should be cited.
3. Line 159: this claim should be cited or supported by data.
4. Line 163: reference to a GEO dataset should include the underlying paper the data originates from.
5. Fig 3 analysis is performed almost entirely in HeLa and HEK293 cells as opposed to the U87-MG cells used in the introduction. As the work is aimed at GBM dynamics, this change in cell type should be mentioned and explained. Other figures where the cell type is not U87-MG should also be identified as such in their descriptions of the results.
6. Fig. 4H suggests that Fsk triggers PKA mediated phosphorylation of Praja2. While Fsk does appear to increase Praja2-mediated ubiquitination of DDX6, its phosphorylation status is not examined in the data.
7. The results of Fig. 7 aim to suggest that praja2 ko strains have increased DDX6-mRNA binding, and that these mRNAs have increased translation. This result may be consistent with the ablation of P-bodies but it is not clear why DDX6 binding would increase mRNA translation, since DDX6 was identified as repressor of translation (PMID: 16179257). The authors should comment on this and provide further support that DDX6 binding is increasing translation.
8. Line 281: How are mRNAs only bound by DDX6 in praja2KO defined? Fig. 7h shows 919 mRNAs while 7j shows 739+23.
9. Line 283-284: The conclusion that praja2 regulates mRNA translation in GBM cells by regulating the binding of specific mRNAs to DDX6 is not sufficiently supported by the data. An overlap of genes does exist but appears modest and needs to be statistically evaluated.
10. Previous studies have suggested that P-body clearance inhibits tumorigenesis and the authors suggest a regulatory role for P-bodies in tumorigenesis in their introduction. How is the accelerated growth of DDX6 allK/R mutants reconciled with this understanding of P-bodies and cancer?

Referee #3:

In this manuscript entitled "An integrated signaling network controls P-bodies dynamics and mRNA translation in glioblastoma" by Senatore et al., the authors report that ubiquitin ligase Praja2, acting downstream of GPCR/cAMP signaling, promotes glioblastoma tumor growth by regulating DDX6-mediated P-body formation. Specifically, they show that Praja2 forms a protein complex with DDX6 and other P-body components and catalyzes non-proteolytic ubiquitination of DDX6 to induce P-body formation. They further show that silencing Praja2 in GBM cells blocks cAMP signaling-induced P-body formation, altering DDX6 interactions with specific mRNAs and impacting translation profiles, ultimately leading to cell senescence and tumor growth arrest in mouse xenograft models.

This study is well-executed and makes interesting and important connections between ubiquitination, P-body dynamics, and GBM tumor growth. It uncovers novel roles of Praja2, including its involvement in DDX6 polyubiquitination and P-body formation relevant to cAMP signaling and mRNA translation. The biochemistry and cellular experiments are elegantly designed with proper controls and well performed, supporting the main conclusions. Nevertheless, there are a few things that could be addressed and clarified to improve the manuscript and its contribution to the field.

1. The study includes multiple RNA sequencing and proteomic experiments. It would be helpful to add supplementary tables describing post-processed data presented in the manuscripts, such as the identified targets from RIP-seq, MS-based PPI analysis, and polysome profiling analysis.
2. Figure 1: It would be helpful to label each panel (A-F) with the relevant datasets. Additionally, ensure panel labels are consistently formatted, either all uppercase or lowercase, to match the text.
3. Figure 2: The names of the identified targets are difficult to read. It would be helpful to increase the font size and provide details, such as names, fold-change, and etc. in a supplemental table for this part.
4. Line 160, a citation is missing for the Yeast-2-hybrid statement.
5. The identified Praja2 interactors also include additional RNA regulatory factors, such as AGO1, EDC4, LSM14A, DCP1. Do these interactions contribute to the observed phenotypes related to P-body nucleation, translation, and tumor growth? Some discussion on this matter would provide additional insights into the topic. It is also interesting that Praja2 does not interact with many other P-body proteins, which seems to be consistent with the observation that Praja2 immunostaining only colocalizes with some P-bodies in cells. Does Praja2 help nucleate some but not all P-bodies?
6. Figure 3E: direct evidence supporting the schematic model, which suggests DCP1A and EDC3 interact with Praja2 indirectly via DDX6, is not clearly presented. Based on their data, it remains unclear whether DCP1A and EDC3 can actually interact directly with Praja2 or not or even are ubiquitinated by Praja2.

7. Is Praja2 alone sufficient to induce P-body formation, or is its activation by the PKA pathway required?
8. Figure 7: The exploration of the exciting findings from polysome profiling and RIP-seq seems inadequate, potentially missing an opportunity to gain more mechanistic insights. For example, what are the identified 515 mRNA (up and down regulated) and 23 targets? How are their functions relevant to GBM? Gene ontology and pathway enrichment analysis might help provide more information on these targets. A supplemental table including details of these identified targets would be helpful.
9. Figure 7H: The authors compared DDX6 RIP vs Input and identified differential binding targets upon Praja2 KO. How does Praja2 KO itself change the transcriptome (Input)?
10. Figure 8G: Adding in quantifications would strengthen the conclusions.
11. Figures 4, 6 and 8 show that Praja2 is required for DDX6 ubiquitination and that Praja2 KO inhibits P-body formation and tumor growth. If the model is correct, DDX6 ubiquitination defective mutant that inhibits P-body formation should also inhibit tumor growth. However, Figure 7 shows that this mutant promotes rather than inhibits tumor growth. Could the authors clarify this discrepancy? A discussion on this and other possible mechanisms would be valuable.
12. It would also be interesting to discuss how the identified DDX6-interacting and translationally regulated mRNA targets in response to Praja2 might be functionally relevant to GBM.
13. The spellings of Praja2/praja2 are not consistent throughout the manuscript. Is there a particular reason for this?

Referee #1:

The authors convincingly show that DDX6 binds to and is ubiquitinated by the Praja2 protein, and this ubiquitination is cAMP dependent. The authors further show that praja2 is necessary for FSK-induced P-body formation and suggest an axis of cAMP-PKA-Praja2-DDX6 in the formation of P-bodies. Interestingly, the authors also show that Praja2 ko can induce premature cellular senescence in glioblastoma cells with in vitro and in vivo evidence. However, the authors describe other findings that disagree with prior research without sufficient comment (DDX6 increasing translation, reduced P-bodies accelerating tumorigenesis). These findings may be highly valuable to the field but would require special attention beyond what is present as they are counterintuitive to past research. An additional serious concern is that there are frequent claims in the initial sections that are made without supporting data or attribution to previous results. I have listed my specific comments below.

R. We wish to thank the reviewer for the positive comments on our work and for the insightful comments that we have carefully addressed as suggested.

Minor Comments

1. P-bodies are not canonically considered organelles (line 80) but may be referred to as membraneless organelles.

R. We have made the suggested change throughout the manuscript.

2. There are grammatical issues throughout the manuscript, most notably for plurals as in the case of P-bodies, (i.e. line 80) "P-bodies dynamics" should be "the dynamics of P-bodies" or "P-body dynamics".

R. Many thanks. We have corrected the grammatical errors.

3. Fig. 2C is a confusing presentation of this data as it applies a linear relationship between cluster A-D. A bar graph would be more suitable.

*R. As suggested, we have now plotted the data of Fig.2C (new **Fig.EV3B**) as bar graph.*

Major Comments

1. The data present in Fig 1 is insufficient for an entire figure as it is only gene ontology analysis. Since it is derived from numerous datasets, efforts should be made to include additional data from the RNA-seq analysis and patient data, not limited to the identity of genes that are regulated beyond just their ontological classification.

*R. Many thanks for this helpful comment. Accordingly, we have now included all the missing information of RNA datasets from control and praja2-depleted GBM cells and substantially revised the whole figure (please, see the **new Fig.1**).*

The presentation of this analysis also should include false discovery rates and enrichment magnitudes instead of just a binary classification as it currently appears to represent.

*R. We thank the reviewer for this comment, The False Discovery Rate has been controlled with the adoption of the Benjamini and Hochberg correction of p-values. We considered "significant" those gene-sets below the threshold of $FDR < 0.01$. We clarified this point in the text (**p.6, line 131-132**). For **Fig.1C**, the enrichment magnitude is associated with the intensity of red and green colors, as commented in the figure legend. We added a color legend to follow the magnitude. For **fig.1D-I**, in the main text of the original submission,*

*this information was omitted to improve the tracking of common gene sets across the experiments. In the **new Table EV1**, the magnitude of enrichment (NES, Normalized Enrichment Score) is listed together with other statistical analysis. To control False positive enrichments, in each analysis we applied the Benjamini and Hochberg correction (**please, see text p7, line 153-155**). Each enrichment is considered significant with a FDR <0.01.*

It is also unclear whether the patient data was performed by the authors or is obtained from publicly available sources; the latter case demands citations alongside those figure notations or in the legend.

R. In the section “Gene-set enrichment analysis on human tissue samples,” we added a reference to each dataset from publicly available sources.

Finally, the conclusion of this section is specious as only a visually suggested similarity between ontological enrichment between patient samples and cell culture samples with siRNA against PRAJA2 is provided.

*R. To support the similarity of enrichment analysis between patient samples and cell cultures, we tested statistical significance of the intersections. For details, please see “Gene-set enrichment analysis on human tissue samples,” section (**pp. 26-27, line 639-661**) and **Appendix Table S1**.*

2. The data used to generate Fig S1A is not cited or discussed. If the authors performed an affinity purification of Praja2, they should show western blot validation. If it is the data of other studies, it should be cited.

R. We have now included this information in the text (p.7, line 167-169) and the appropriate reference (Delle Donne et al, 2022; Rinaldi et al, 2024).

3. Line 159: this claim should be cited or supported by data.

R. We have now updated this information by including the missing reference (Lignitto et al, 2011)

4. Line 163: reference to a GEO dataset should include the underlying paper the data originates from.

R. We have now added the appropriate reference (doi.org/10.1101/675439).

5. Fig 3 analysis is performed almost entirely in HeLa and HEK293 cells as opposed to the U87-MG cells used in the introduction. As the work is aimed at GBM dynamics, this change in cell type should be mentioned and explained. Other figures where the cell type is not U87-MG should also be identified as such in their descriptions of the results.

*R. We have now added praja2/DDX6 co-localization also in U87MG (Please, see **new Fig.3C**). Co-immunoprecipitation of endogenous praja2/DDX6 complex, shown in the new Fig.3D, was performed in U87MG cells, while Co-IP experiments showing the binding between wild type and mutant proteins were performed in transfected HEK293 cells, a very efficient cell system to overexpress exogenous proteins. The binding between praja2 and DDX6 bodies has been confirmed also in GBM cells.*

6. Fig. 4H suggests that Fsk triggers PKA mediated phosphorylation of Praja2. While Fsk does appear to increase Praja2-mediated ubiquitination of DDX6, its phosphorylation status is not examined in the data.

R. *Praja2 is activated by PKA phosphorylation and effectively we demonstrated that this event is required for DDX6 ubiquitylation mediated by praja2 (Fig.4A, Fig.4D).*

7. The results of Fig. 7 aim to suggest that praja2 ko strains have increased DDX6-mRNA binding, and that these mRNAs have increased translation. This result may be consistent with the ablation of P-bodies but it is not clear why DDX6 binding would increase mRNA translation, since DDX6 was identified as repressor of translation (PMID: 16179257). The authors should comment on this and provide further support that DDX6 binding is increasing translation.

R. *We demonstrated that in absence of praja2 there are more mRNAs bound to DDX6 and protein translation is upregulated. We found only 23 transcripts bound to DDX6 in praja2KO cells that are translationally regulated by praja2, indicating that the control of overall protein translation depends minimally by praja2-mediated regulation of P-bodies and DDX6-mRNAs complexes. DDX6 is also localized at ribosomes (<https://doi.org/10.7554/eLife.92426.2>; <https://doi.org/10.1016/j.molcel.2015.08.014>), suggesting that a portion of mRNAs bound to DDX6 identified in RIP experiments can be located at polysomes.*

8. Line 281: How are mRNAs only bound by DDX6 in praja2KO defined? Fig. 7h shows 919 mRNAs while 7j shows 739+23.

R. *As suggested, we have corrected the data shown in the Fig.7H.*

9. Line 283-284: The conclusion that praja2 regulates mRNA translation in GBM cells by regulating the binding of specific mRNAs to DDX6 is not sufficiently supported by the data. An overlap of genes does exist but appears modest and needs to be statistically evaluated.

R. *Results from RIP analysis and polysome profiling have been obtained from 3 biological replicates and a statistical analysis of the difference among transcripts have been adjusted with a p value <0.05. Our data indicate that praja2 controls the overall protein translation in GBM cells that is only in part dependent on DDX6/mRNAs binding. We have now better explained this point in the main text (Please, see p.14, line 339-342)*

10. Previous studies have suggested that P-body clearance inhibits tumorigenesis and the authors suggest a regulatory role for P-bodies in tumorigenesis in their introduction. How is the accelerated growth of DDX6 allK/R mutants reconciled with this understanding of P-bodies and cancer?

R. *We wish to thank the reviewer to point this important aspect of the role of P body clearance in tumorigenesis. Looking at published literature on this topic, the role of P bodies in cancer biology is still controversial. There are papers indicating a positive role of P bodies clearance in tumor growth (see 10.15252/embr.202050835), while others suggests that P bodies formation supports tumor growth (10.1038/s44318-024-00120-6). We believe that the dynamic nature of these condensates can exert apparently opposite effects on tumorigenesis, acting as main orchestrators of cancer cell plasticity and playing a dual role in cancer cell growth or drug resistance in growth-arrested cancer cells, conferring resilience and plasticity to cancer.*

Referee #3:

In this manuscript entitled "An integrated signaling network controls P-bodies dynamics and mRNA translation in glioblastoma" by Senatore et al., the authors report that ubiquitin ligase Praja2, acting downstream of GPCR/cAMP signaling, promotes glioblastoma tumor growth by regulating DDX6-mediated P-body formation. Specifically, they show that Praja2 forms a protein complex with DDX6 and other P-body components and catalyzes non-proteolytic ubiquitination of DDX6 to induce P-body formation. They further show that silencing Praja2 in GBM cells blocks cAMP signaling-induced P-body formation, altering DDX6 interactions with specific mRNAs and impacting translation profiles, ultimately leading to cell senescence and tumor growth arrest in mouse xenograft models.

This study is well-executed and makes interesting and important connections between ubiquitination, P-body dynamics, and GBM tumor growth. It uncovers novel roles of Praja2, including its involvement in DDX6 polyubiquitination and P-body formation relevant to cAMP signaling and mRNA translation. The biochemistry and cellular experiments are elegantly designed with proper controls and well performed, supporting the main conclusions. Nevertheless, there are a few things that could be addressed and clarified to improve the manuscript and its contribution to the field.

R. Many thanks to the reviewer in finding our manuscript well-executed, elegantly designed with proper controls and well performed, and for insightful suggestions and comments.

1. The study includes multiple RNA sequencing and proteomic experiments. It would be helpful to add supplementary tables describing post-processed data presented in the manuscripts, such as the identified targets from RIP-seq, MS-based PPI analysis, and polysome profiling analysis.

*R. As suggested, we have now added extended view (EV) tables for RIP experiments, polysome profiling, RNA sequencing and proteomic analysis. Please, see new **EV table 1-8**.*

2. Figure 1: It would be helpful to label each panel (A-F) with the relevant datasets. Additionally, ensure panel labels are consistently formatted, either all uppercase or lowercase, to match the text.

*R. We have now included in the new **Fig.1** the name and details of the datasets.*

3. Figure 2: The names of the identified targets are difficult to read. It would be helpful to increase the font size and provide details, such as names, fold-change, and etc. in a supplemental table for this part.

*R. Thanks for this comment. We have now modified the network shown in the new **Fig.2A**, have increased the font size and added a graph showing the FDR (new **Fig.2B**). A supplementary table with all the interactors identified has been included (please, see **Table EV2**).*

4. Line 160, a citation is missing for the Yeast-2-hybrid statement.

R. We have Included the appropriate reference (Lignitto et al 2011).

5. The identified Praja2 interactors also include additional RNA regulatory factors, such as AGO1, EDC4, LSM14A, DCP1. Do these interactions contribute to the observed phenotypes related to P-body nucleation, translation, and tumor growth? Some discussion on this matter would provide additional insights into the topic. It is also interesting that Praja2 does not interact with many other P-body proteins, which seems to be consistent with the observation that Praja2 immunostaining only colocalizes with some P-bodies in cells. Does Praja2 help nucleate some but not all P-bodies?

R. We thank the Reviewer to raise this important point. Published work indicate that praja2 localizes at different cellular compartments, including centrosome, primary cilium, cellular membrane, Golgi and ER membranes, where it acts as a major signaling relay controlling protein stability and activity. Here, we show that praja2 staining only partly colocalizes with P-bodies, suggesting that praja2 binding to DDX6 and localization of the complex at P-bodies is a dynamic process regulated by signaling events in response to specific cellular needs or extracellular stimuli. The finding that praja2 staining only colocalizes with some P-bodies suggests that praja2 binding to DDX6 and its localization at P-bodies is a dynamic process possibly regulated by signaling events in response to specific cellular needs or extracellular stimuli. The mechanism operated by praja2 may also involve other components of P-bodies, including EDC4, LSM14A, DCP1, with important implications for organelle biology and dynamics. Nevertheless, the findings that deletion of praja2 or expression of a DDX6 ubiquitylation-defective mutant decreases the number of P-bodies indicate that PKA-praja2-DDX6 axis is, indeed, a relevant mechanism for the organelle assembly. As suggested, we have now included a brief discussion on this topic in the manuscript (please, see p.18, line 430-437)

6. Figure 3E: direct evidence supporting the schematic model, which suggests DCP1A and EDC3 interact with Praja2 indirectly via DDX6, is not clearly presented. Based on their data, it remains unclear whether DCP1A and EDC3 can actually interact directly with Praja2 or not or even are ubiquitinated by Praja2.

R. Many thanks for this comment. Our data indicate that praja2 directly interacts with DDX6 and praja2/DDX6 complex contains also EDC3 and DCP1A proteins. However, as pointed by the reviewer, we have no information about how the whole complex is formed. Accordingly, we have presented a schematic model focusing on the direct binding between DDX6 and praja2 (Fig.3F).

7. Is Praja2 alone sufficient to induce P-body formation, or is its activation by the PKA pathway required?

R. This is an important point. Our data indicate that praja2 alone is sufficient to increase p bodies number (from Fig.5I-J), as well as DDX6 ubiquitylation (from Fig.4A). To further support the role of PKA-praja2 axis in the control of P-body assembly, we analyzed P-body formation in cells expressing a praja2 mutant that cannot be phosphorylated by PKA. This mutant carries a double mutation at the PKA phosphorylation sites ser342 e thr389)(praja2DM) (Lignitto et al.,2011). As shown in the new Fig.5I-J, expression of praja2 significantly increased the number of P-bodies in unstimulated cells, compared to wild-type cells and cells expressing mutant praja2 (praja2DM). Stimulation with Forskolin increased the number of P-bodies in control cells, while had no major effects on praja2DM-expressing cells (Figure 5I-J). These data clearly indicate that PKA phosphorylation of praja2 is, indeed, required for P-body assembly.

8. Figure 7: The exploration of the exciting findings from polysome profiling and RIP-seq

seems inadequate, potentially missing an opportunity to gain more mechanistic insights. For example, what are the identified 515 mRNA (up and down regulated) and 23 targets? How are their functions relevant to GBM? Gene ontology and pathway enrichment analysis might help provide more information on these targets. A supplemental table including details of these identified targets would be helpful.

R. *As suggested by the reviewer, we performed a gene ontology analysis of biological processes and enriched pathway of the 23 identified transcripts (Fig.7K) and highlighted a correlation between some of these processes/pathways and GBM (pp.13-14, line 324-342). We have also included tables containing the identified transcripts and the relevant information of RIP seq and polysome profiling analysis (Table EV4,7-10).*

9. Figure 7H: The authors compared DDX6 RIP vs Input and identified differential binding targets upon Praja2 KO. How does Praja2 KO itself change the transcriptome (Input)?

R. *We have now included a differential expression analysis of transcripts (input) between praja2KO and wild-type GBM cells. As shown in the new Fig.EV5 and new Table EV5-6, praja2 deletion had major effects on the transcriptome of GBM cells, as observed also in praja2-silenced GBM cells (please, see Fig.3A-C).*

10. Figure 8G: Adding in quantifications would strengthen the conclusions.

R. *We have performed the quantitative analysis of Figure 8G. The data are shown in the new Appendix Fig.S4.*

11. Figures 4, 6 and 8 show that Praja2 is required for DDX6 ubiquitination and that Praja2 KO inhibits P-body formation and tumor growth. If the model is correct, DDX6 ubiquitination defective mutant that inhibits P-body formation should also inhibit tumor growth. However, Figure 7 shows that this mutant promotes rather than inhibits tumor growth. Could the authors clarify this discrepancy? A discussion on this and other possible mechanisms would be valuable.

R. *Many thanks to the reviewer for this important comment. The findings reported in the manuscript, in line with previous work, demonstrated that praja2 knockout markedly affects cell proliferation and tumor growth. At mechanistic level, praja2 KO deregulates the stability and activity of signaling enzymes underlying oncogenic (Ras-KSR1) and oncosuppressive (MOB1/LATS1) pathways, cancer metabolic rewiring (KSR2-AMPK) and cell survival. Here, we report that praja2 ubiquitylates DDX6 at multiple lysin residues and the expression of a DDX6 ubiquitylation mutant (allK/R) accelerates tumor growth. The apparent discrepancy between the inhibitory effects of praja2 KO and those observed expressing the DDX6 ubiquitylation-mutant on tumor growth might be explained by considering the more general role of praja2 in tumor cell biology (global translation, mitogenesis, metabolism, survival and growth control) that cannot be recapitulated by DDX6 allK/R mutant. (Please, see p.21, line 498-502)*

12. It would also be interesting to discuss how the identified DDX6-interacting and translationally regulated mRNA targets in response to Praja2 might be functionally relevant to GBM.

R. *Please, see response to point 8.*

13. The spellings of Praja2/praja2 are not consistent throughout the manuscript. Is there a particular reason for this?

R. *Corrected*

Dear Prof. Feliciello

Thank you for the submission of your revised manuscript to EMBO reports. We have now received the full set of referee reports that is copied below.

As you will see, both referees are very positive about the study and request only minor changes to clarify text and figures.

From the editorial side, there are also a few things that we need before we can proceed with the official acceptance of your study.

- Please provide up to 5 keywords.

- Please change the header "Availability of data and materials." To "Data Availability". Please remove the following statement from the Data availability paragraph: "All other relevant data and materials are available from the corresponding author on reasonable request." This paragraph should only contain references to datasets in public repositories (and source data). Please be reminded to remove the reviewer access data.

- Please update the 'Conflict of interest' paragraph to our new 'Disclosure and competing interests statement', place it after the Acknowledgments and remove it from the title page. For more information see <https://www.embopress.org/page/journal/14693178/authorguide#conflictsofinterest>

- Please resolve the following author name discrepancy: Antonietta Arcella in the manuscript vs. Antonella Arcella in the online manuscript tracking system.

- Regarding the Author Contributions, we now use CRediT to specify the contributions of each author in the journal submission system. Therefore, please remove the Author Contributions from the manuscript file and make sure that the author contributions in our online manuscript tracking system are correct and up-to-date. The information you specified in the system will be automatically retrieved and typeset into the article. You can enter additional information in the free text box provided, if you wish.

- I noticed that co-author Giorgio Giurato has deceased. I am very sorry to hear this but in light of this information, I would ask you to reconsider authorship. Since Mr. Giurato might not have had the possibility to contribute to the writing of the study and give his approval of the final version, it might be advisable to list his contribution in the Acknowledgments rather than in the author list. I am more than happy to discuss this further, if you wish.

- Funding:

Information on funding needs to be part of Acknowledgments.

PRIN2022 is missing in the online manuscript tracking system and needs to be added.

The following items may need to be entered in the online manuscript tracking system via the 'More Funders' option: IBISA, the Cancéropôle PACA, the Provence-Alpes-Côte d'Azur Region, the Institut Paoli-Calmettes, and Fonds Européen de Développement Regional (FEDER) .

- EV Tables:

a) You have currently 10 EV tables, most of which are complex datasets and should therefore be uploaded as file type "Dataset" and the name changed to "Dataset EV#".

b) Table EV8 could remain a "Table", it should be called Table EV1.

c) Please update the source file names, their title in the online submission system and all callouts in the manuscript accordingly.

d) Please add the Dataset or table name in the first tab of all .xls files, as a header of the legend. Please also add it to the .txt files.

- You refer to a Reagents and Resources table in the Author Checklist but it seems that the table is missing. Please download and fill our Reagents and Tools Table template (.docx), which you can find in our author guidelines:

Appendix Table S2 (Real time primers) must be incorporated in the R&T table as well.

- Please add page numbers in the Appendix (also to the table of content on the first page).

- Appendix figure S4: would it be more appropriate to calculate the mean per mouse and then derive statistics from the 'mean of

the means". Taking all cells as independent samples into the statistical analysis seems to represent pseudoreplication? But I might be mistaken. Can you please add how many cells you have counted from how many mice? Please also note that we ask to specify the exact p-values in the figure legends.

- Appendix Table S1 and data citations: you refer to published datasets in the legend of this table. Please use our citation format for datasets (data citations). See the relevant section "Data citation" in our guide to authors and the information in the Author Checklist, C115.

The same applies to the relevant section in the Methods, which refers to the same datasets:

"Two scRNA-seq (GSE129671, (Ding et al, 2019) 701 and 702) data sets, the GBM bulk RNA-seq study from TCGA, and the proteomic GBM profiles (Proteomic Data Common, ID: PDC000204 (Wang et al, 2021) were analyzed. Additionally, we used scRNA-seq data (GSE67835 (Darmanis et al, 2015) from healthy adult brain donors."

Please cite the study that reported these datasets, if applicable, and then provide a second 'data citation' with a link that directly resolves to the datasets you have reused. The latter is labeled 'Data ref' in the text and with [DATASET] in the references.

A short description of data citations for your reference: Our journal encourages inclusion of *data citations in the reference list* to directly cite datasets that were re-used and obtained from public databases. Data citations in the article text are distinct from normal bibliographical citations and should directly link to the database records from which the data can be accessed. In the main text, data citations are formatted as follows: "Data ref: Smith et al, 2001" or "Data ref: NCBI Sequence Read Archive PRJNA342805, 2017". In the Reference list, data citations must be labeled with "[DATASET]". A data reference must provide the database name, accession number/identifiers and a resolvable link to the landing page from which the data can be accessed at the end of the reference. Further instructions are available at <<https://www.embopress.org/page/journal/14693178/authorguide#referencesformat>>.

- Author checklist, C96: you indicate that you used specimens and field samples. Please check whether this answer indeed applies, as it seems that you used only cultured cells and mice.

- Ethical approval: Please add the authority granting the approval in the text.

- Mouse work: please describe the housing and husbandry conditions and the genetic background, sex, and age of the mice.

- Figure 9B: the data indicate that the tumor volume reached 1500 mm³ at the end point. For animals carrying a single tumor, the mean diameter should normally not exceed 1.2 cm in mice, which would equal approx. 900 mm³ tumor volume. That said, the tumor size would fall within the 10% of body weight limit. Can you please define, which criteria were used to determine the endpoint?

- Ding et al, 2019 is a preprint. It is correctly labeled in the reference list, but needs the prefix 'preprint:' in the in-text citation.

- The Abbreviations section needs to be removed from the manuscript. Abbreviations should be defined in brackets after their first mention in the text, not in a list of abbreviations.

- Methods and protocols should be Methods.

- Please remove the methods from the Appendix and include them in the main manuscript methods section.

- The gene names in Figure 1A are partially overlapping and therefore difficult to discern.

- Our production/data editors have asked you to clarify several points in the figure legends (see below). Please incorporate these changes in the manuscript and return the revised file with tracked changes with your final manuscript submission.

A) Statistical test information. Only p-values that are actually shown in the figure panel(s) should (and must) be defined in the legends, all others should be removed from (or added to) the legend. Moreover, we ask for the specification of exact p-values:

- Please note that the exact p values are not provided in the legends of figures 5B, C, E, F, H, J; 6B, E, F; 7E, G, I; 8B, D, F; 9B, C, D; EV4 B, C; supplementary figure(s) 4A-C.

- Please indicate the statistical test used for data analysis in the legends of figures 1A, 7C, K, EV5 A, B; supplementary figure(s) 2B-D

- Please indicate what */ **/ ***/ **** represents; if this represents p value(s), please specify exact p value in the legend(s) of figure(s) 7B"

B) Replicates and error bars:

- Please note that information related to n is missing in the legends of figures 1A, 7C, 8F; EV3 B, EV5 A; supplementary figure(s) 2B-D

- Although 'n' is provided, please describe the nature of entity for 'n' in the legends of figures 7B

- Please note that the error bars are not defined in the legends of figures 7I. "

D) Data presentation:

- Please note that scale bar and its definition are missing for figure 8A

- When doing a spotcheck on the source data I noticed that the praja2 image for Figure 3C is actually a merge with the DAPI channel. Please check this one.

- As a standard procedure we edit the title and abstract of manuscripts to make them more accessible to our general readership. Please find my suggestion below my signature.

- Finally, EMBO Reports papers are accompanied online by

A) a short (1-2 sentences) summary of the findings and their significance,

B) 2-3 bullet points highlighting key results and

C) a schematic summary figure that provides a sketch of the major findings (not a data image).

Please provide the summary figure as a separate file in PNG or JPG format at a size of 550x300-600 pixels (width x height).

Please note that the size is rather small and that text needs to be readable at the final size. Please send us this information along with the revised manuscript.

With kind regards,

=====

Referee #1:

The authors have performed additional experiments and analysis that have resolved most of my comments. Overall, the manuscript has improved substantially, and the conclusions are better supported.

Minor point:

The authors appear to use the word "transcriptional" where it seems "transcriptomic" is the intended meaning, since the authors don't actually measure transcription. For example, lines 126, 128 and elsewhere.

Referee #3:

The authors have now addressed all my questions, and the manuscript is suitable for publication.

=====

Praja2 promotes P-body assembly and translation in glioblastoma by non-proteolytic ubiquitination of DDX6

Glioblastoma multiforme (GBM) is the most lethal form of malignant brain tumor in adults. Dysregulation of protein synthesis contributes to cancer cell plasticity, driving GBM cell heterogeneity, metastatic behavior and drug resistance. Understanding the complex network and signaling pathways governing protein translation, is therefore an important goal for GBM treatment. Here we identify a novel signaling network centered on the E3 ubiquitin ligase praja2 that controls protein translation in GBM. Praja2 forms a multimeric complex with the RNA helicase DDX6, which inhibits translation of target RNAs within processing bodies (P-bodies). Stimulation of cAMP signaling through activation of G protein-coupled receptors induces P-body assembly through praja2-mediated non-proteolytic polyubiquitylation of DDX6. Genetic inactivation of praja2 reshapes DDX6/mRNA complexes and translating polysomes and promotes cellular senescence and GBM growth arrest. Expression of an ubiquitylation-defective DDX6 mutant suppresses assembly of P-bodies and sustains GBM growth. Taken together, our findings identify a cAMP-driven network that controls translation in P-bodies and GBM growth.

All editorial and formatting issues were resolved by the authors.

Prof. Antonio Feliciello
University Federico II
Biologia e Patologia Cellulare e Molecolare
via S. Pansini, 5
Napoli 80131
Italy

Dear Prof. Feliciello,

I am very pleased to accept your manuscript for publication in the next available issue of EMBO reports. Thank you for your contribution to our journal.

Yours sincerely,
